# Neural dynamics at successive stages of the ventral visual stream are consistent with hierarchical error signals

**Elias B Issa[†]\*, Charles F Cadieu[‡], James J DiCarlo**

Department of Brain and Cognitive Sciences, McGovern Institute for Brain Research, Massachusetts Institute of Technology, Cambridge, United States

**Abstract** Ventral visual stream neural responses are dynamic, even for static image presentations. However, dynamical neural models of visual cortex are lacking as most progress has been made modeling static, time-averaged responses. Here, we studied population neural dynamics during face detection across three cortical processing stages. Remarkably,~30 milliseconds after the initially evoked response, we found that neurons in intermediate level areas decreased their responses to typical configurations of their preferred face parts relative to their response for atypical configurations even while neurons in higher areas achieved and maintained a preference for typical configurations. These hierarchical neural dynamics were inconsistent with standard feedforward circuits. Rather, recurrent models computing prediction errors between stages captured the observed temporal signatures. This model of neural dynamics, which simply augments the standard feedforward model of online vision, suggests that neural responses to static images may encode top-down prediction errors in addition to bottom-up feature estimates.
DOI: https://doi.org/10.7554/eLife.42870.001

\*For correspondence:
elias.issa@columbia.edu

Present address: [†]Department of Neuroscience, Zuckerman Mind Brain Behavior Institute, Columbia University, New York, United States; [‡]Bay Labs, Inc, San Francisco, United States

Competing interests: The authors declare that no competing interests exist.

## Introduction

The primate ventral visual stream is a hierarchically organized set of cortical areas beginning with the primary visual cortex (V1) and culminating with distributed patterns of neural firing across the inferior temporal cortex (IT) that explicitly encode objects (i.e. linearly decodable object identity) (*Hung et al., 2005*) and quantitatively account for core invariant object discrimination behavior in primates (*Majaj et al., 2015*). Formalizing object recognition as the result of a series of feedforward computations yields models that achieve impressive performance in object categorization (*Krizhevsky et al., 2012*; *Zeiler and Fergus, 2013*) similar to the absolute level of performance achieved by IT neural populations, and these models are the current best predictors of neural responses in IT cortex and its primary input layer, V4 (*Cadieu et al., 2014*; *Yamins et al., 2014*). Thus, the feedforward inference perspective provides a simple but powerful, first-order framework for the ventral stream and core invariant object recognition.

However, visual object recognition behavior may not be executed via a single feedforward neural processing pass (a.k.a. feedforward inference) because IT neural responses are well-known to be dynamic even in response to images without dynamic content (*Brincat and Connor, 2006*; *Sugase et al., 1999*; *Chen et al., 2014*; *Meyer et al., 2014*), raising the question of what computations those neural activity dynamics might reflect. Prior work has proposed that such neuronal response dynamics could be the result of different types of circuits executing different types of computation such as: (1) recurrent circuits within each ventral stream processing stage implementing local normalization of the feedforward information as it passes through that stage (*Carandini et al., 1997*; *Schwartz and Simoncelli, 2001*; *Carandini and Heeger, 2011*), (2) feedback circuits between each pair of ventral stream stages implementing the integration of top-down with bottom-up

information to improve the current (online) inference (*Seung, 1997*; *Lee et al., 2002*; *Zhang and von der Heydt, 2010*; *Epshtein et al., 2008*), or (3) feedback circuits between each pair of stages comparing top-down and bottom-up information to compute prediction errors that guide changes in synaptic weights so that neurons are better tuned to features useful for future feedforward behavior (learning) (*Rao and Ballard, 1999*). Thus, neural dynamics may reflect the various adaptive computations (within-stage normalization, top-down Bayesian inference) or reflect the underlying error intermediates that could be generated during those processes (e.g. predictive coding).

These computationally motivated ideas can each be formalized in neural circuits that contain feedforward, lateral (normalization), or feedback (hierarchical Bayesian inference) connections to ask which connection motif best predicts response dynamics across the visual hierarchy. Here, our main goal was to look beyond the initial, feedforward response edge to see if we could disambiguate among dynamics that might result from stacked feedforward, lateral, and feedback operations. Rather than record from a single processing level, we measured the dynamics of neural signals across three hierarchical levels (pIT, cIT, aIT) within macaque IT. We focused on face processing subregions within each of these levels for three reasons. First, prior evidence argues that these three face processing subregions are tightly anatomically and functionally connected and that the subregion in pIT is the dominant input to the higher subregions (*Grimaldi et al., 2016*; *Moeller et al., 2008*). Second, because prior work argues that a key behavioral function of these three subregions is to distinguish faces from non-faces, this allowed us to focus our testing on a relatively small number of images targeted to engage that processing function. Third, prior knowledge of pIT neural tuning properties (*Issa and DiCarlo, 2012*) allowed us to design images that were quantitatively matched in their ability to drive neurons in the pIT input subregion but that should ultimately be processed into two separate groups (face vs non-face). We reasoned that these images would force important computations for disambiguation to occur somewhere between the pIT subregion and the higher level (cIT, aIT) subregions. With this setup, our aim was to observe the dynamics at all three levels of the hierarchy in response to that image processing challenge so that we might discover – or at least constrain – which type of computation is at work.

Consistent with the idea that the overall system performs – among other things – face vs non-face discrimination (i.e. face detection), we found that in the highest face processing stage (aIT), neurons rapidly developed and maintained a response preference for the typical frontal configuration of the face parts even though our images were designed to be challenging for frontal face detection. However, we found that many neurons in the early (pIT) and intermediate (cIT) processing levels of IT paradoxically showed an overall stronger response for atypical face-part configurations relative to typical face-part configurations over time. That is, these neurons evolved an apparent preference for images of misarranged face parts within 30 milliseconds of their feedforward response. We found that standard feedforward models that employ local recurrences such as adaptation, lateral inhibition, and normalization could not capture this stage-wise pattern of image selectivity despite our best attempts. However, we found that a *decreasing* – rather than increasing – relative preference for typical face-part configurations in early and intermediate processing stages is a natural dynamical signature of previously suggested 'error coding' models (*Rao and Ballard, 1999*) in which the neural spiking activity at each processing stage carries both an explicit representation of the variables of interest (e.g. Is an eye present? And is a whole face present?) and an explicit encoding of errors computed between each pair of stages in the hierarchy (e.g. a face was present, but the eye was not present at the correct location).

## Results

We leveraged the hierarchically arranged face processing system in macaque ventral visual cortex to study the dynamics of neural processing across a hierarchy (*Tsao et al., 2006*; *Tsao et al., 2008*) (*Figure 1A*). The serially arranged posterior, central, and anterior face-selective subregions of IT (pIT, cIT, and aIT) can be conceptualized as building increasing selectivity for faces culminating in aIT representations (*Freiwald and Tsao, 2010*; *Chang and Tsao, 2017*). Using serial, single electrode recording, we sampled neural sites across the posterior to anterior extent of the IT hierarchy in the left hemispheres of two monkeys to generate neurophysiological maps (*Figure 1A*; example neurophysiological map in one monkey using a faces versus non-face objects screen set) (*Issa et al., 2013*). We localized the recording locations in vivo and co-registered across all penetrations using a

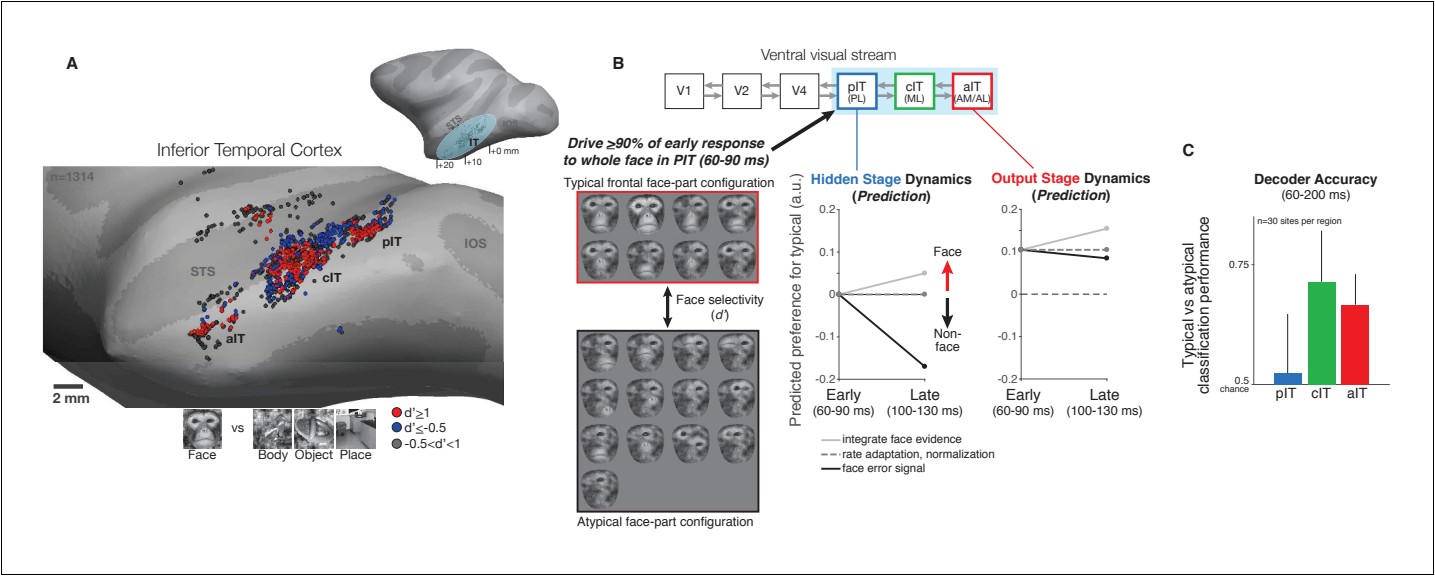

**Figure 1.** Neural recordings and experimental design in face-selective subregions of the ventral visual stream. (**A**) Neurons were recorded along the lateral convexity of the inferior temporal lobe spanning the posterior to anterior extent of IT (+0 to+20 mm AP, Horsely-Clarke coordinates) in two monkeys (data from monkey one are shown). Based on prior work, face-selective sites (red) were operationally defined as those with a response preference for images of frontal faces versus images of non-face objects (see Materials and methods). While these neurons were found throughout IT, they tended to be found in clusters that mapped to previously identified subdivisions of IT (posterior, central, and anterior IT) and corresponded to face-selective areas identified under fMRI in the same subjects (*Issa and DiCarlo, 2012*; *Issa et al., 2013*) (STS = superior temporal sulcus, IOS = inferior occipital sulcus, OTS = occipitotemporal sulcus). (**B**) (top diagram) The three visual processing stages in IT lie downstream of early visual areas V1, V2, and V4 in the ventral visual stream. (left) We designed our stimuli to focus on the intermediate stage pIT by seeking images of faces and images of non-faces that would, on average, drive equally strong initial responses in pIT. Novel images were generated from an exemplar monkey face by positioning the face parts in different positions within the face outline. This procedure generated both frontal face and non-face arrangements of the face parts, and we identified 21 images (red and black boxes) that drove the mean, early (60–100 ms) pIT population response to $\geq$ 90% of its response to the intact face (first image in red box is synthesized whole face; compare to the second image which is the original whole face), and of these 21 images, 13 images contained atypical, non-face arrangements of the face parts. For example, images with an eye centered in the outline (black box, 3$^{rd}$ and 4$^{th}$ rows) as opposed to the lateralized position of the eye in a frontal face (red box) have a global interpretation ('cyclops') that is not consistent with a frontal face but still evoked strong pIT responses. Selectivity of neural sites (see *Figure 3 and 4*) for typical versus atypical face-part configuration images was quantified using a d' measure. (middle) Computational hypotheses of cortical dynamics make differing predictions about how neural selectivity in pIT may evolve following images with similar local face features matched in their ability to evoke initial response but with different spatial context (typical vs atypical part configuration of the face). (right) Predictions of how aIT would behave as an output stage building selectivity for images of with face parts configured in the typical frontal face configuration through multiple stages of processing. (**C**) A population decoder, trained on average firing rates (60–200 ms post image onset, linear SVM classifier) for typical frontal face versus atypical part configurations of the face parts in this image subset, performed poorly in pIT on held-out trials of the same images (trial splits used so that the same images were shown in classifier training (90% of trials) and testing (10% of trials)). However, the particular configuration (typical vs atypical) could be determined at above chance levels when reading the cIT and aIT population responses.

DOI: https://doi.org/10.7554/eLife.42870.002

stereo microfocal x-ray system (~400 micron in vivo resolution) (*Cox et al., 2008*; *Issa et al., 2010*) allowing accurate assignment of sites to different face processing stages (n = 633 out of 1891 total sites recorded were assigned as belonging to a face-selective subregion based on their spatial location; see Materials and methods). Results are reported here for sites that were spatially located in a face-selective subregion, that showed visual drive to any category in the screen set (see Materials and methods), and that were subsequently tested with our face versus non-face challenge set (*Figure 1B*, left panel) (n = 115 pIT, 70 cIT, and 40 aIT sites).

Our experimental design was intended to test previously proposed computational hypotheses of hierarchical neural dynamics during visual face processing (*Figure 1B*). Briefly, these hypotheses predict how stimulus preference (in this instance, for typical versus atypical configurations of the face parts) might change over time in a neural population (*Figure 1B*, middle panel): (1) simple spike-rate adaptation predicts that initial rank-order selectivity (i.e. relative stimulus preference) will be largely preserved (*Figure 1B*, dashed line) while neurons adapt their absolute response strength over time,

(2) local normalization predicts that stronger responses are in some cases normalized to match weaker responses based on population activity to specific dimensions (*Carandini et al., 1997*); importantly, normalization is strongest for nuisance (non-coding) dimensions (e.g. low versus high stimulus contrast) and in its idealized form would not alter selectivity along coding dimensions (e.g. typical versus scrambled feature configurations) (*Figure 1B*, dashed line), (3) evidence accumulation through temporal integration, winner-take-all through recurrent inhibition, or Bayesian inference through top-down feedback mechanisms all qualitatively predict a similar or stronger response over time for preferred features presented in the learned, typical face-part configuration versus presentation in an unexperienced atypical face-part configuration (*Lee and Mumford, 2003*) (*Figure 1B*, light gray line), and (4) predictive coding posits that, for neurons that are coding error, their responses would show the opposite trend being greater for stimuli containing their preferred features but presented in configurations inconsistent with predictions of a typical frontal face (*Rao and Ballard, 1999*) (*Figure 1B*, black line). Note, that error signaling is a qualitatively different computation than normalization, as error coding predicts a decreased response along the coding dimension (typical vs atypical configuration of features) whereas normalization would ideally not affect selectivity for typical versus atypical face-part configurations and only affect variation along orthogonal, nuisance dimensions. Properly testing these predictions (no change, increased, or decreased response over time to preferred face parts presented in typical versus atypical configurations) requires measurements from the intermediate stages of the hierarchy as all of these models operate under the premise that the system builds and maintains a preference for typically configured face parts at the top of the hierarchy (*Figure 1B*, right, and see Introduction). Thus, the intermediate stages (here pIT, see *Figure 1B*) are most likely to be susceptible to confusions from typical/atypical face-part configurations and thus be influenced by, for example, the top-down mechanisms posited in Bayesian inference and predictive coding where higher areas encoding faces generate predictions of the face features and their locations that directly influence the responses of lower areas encoding those local face features (*Lee and Mumford, 2003*; *Rao and Ballard, 1999*).

To ensure that we are observing purely visual predictions and not signals from other downstream top–down mechanisms, it is important to consider the effects of potential arousal and attention signals. To limit the effect of arousal to surprising novel face-part configurations, we presented atypical stimuli rapidly (100 ms on, 100 ms off) and in a randomly interleaved fashion with typical stimuli. Given that endogenous attention mechanisms operate on timescales of hundreds of milliseconds (*Ward et al., 1996*; *Müller et al., 1998*; *Egeth and Yantis, 1997*) and that the preceding stimulus is not predictive of the next, dynamics observed during the first hundred milliseconds of the response are likely not the result of neural mechanisms that are hypothesized to be at work in endogenous attention.

## Typical and atypical configurations of face parts driving similar initial responses in pIT

Here, we chose to focus our key, controlled tests on pIT – an intermediate stage in the ventral stream hierarchy, but the first stage within IT where neural specialization for face detection (i.e. face vs non-face) has been reported (*Grimaldi et al., 2016*). Consistent with its intermediate position in the ventral visual system, we had previously found that pIT face-selective neurons are not truly selective for whole faces but respond to local face features, specifically those in the eye region (*Issa and DiCarlo, 2012*). Taking advantage of this prior result, we created face stimuli and similar but non-facelike stimuli that were customized to challenge the face processing system in that each would strongly drive pIT responses, thus forcing the higher IT stages to complete the discrimination between face and challenging non-face images based on higher-level information. To generate these challenging images, we systematically varied the positions of parts, in particular the eye, within the face (*Issa and DiCarlo, 2012*) (see Materials and methods). This set included images that contained face parts in positions consistent with a frontal view of a face or images that only differed in the relative spatial configuration of the face parts within the face outline (*Figure 1B*, left). Of the 82 images screened, we identified 21 part configurations that each drove the pIT population response to ≥90% of its response to a correctly configured whole face. Of those 21 images, 13 images were inconsistent with the face-part configuration of a frontal face (*Figure 1B*, black box). For the majority of the results that follow, we focus on comparing the neural responses to these 13 pIT-matched images that could *not* have arisen from frontal faces (referred to hereafter as 'atypical face-part

configurations') with the 8 images that could have arisen from frontal faces (referred to hereafter as 'typical face-part configurations'). Again, we stress that these two groups of images were selected to evoke closely matched initial pIT population activity.

Importantly, the pIT-matched images used here presented a more stringent test of face vs non-face discrimination than prior work. Specifically, most prior work used images of faces and non-face objects ('classic images') that contain differences across multiple dimensions including local contrast, spatial frequency, and types of features (*Tsao et al., 2006*; *Afraz et al., 2006*; *Moeller et al., 2017*; *Sadagopan et al., 2017*). Consistent with this, we found that the population decoding classification accuracy of our recorded neural populations using these classic images (faces versus non-face objects) is near perfect (>99% in pIT, cIT, and aIT, n = 30 sites per region). However, we found that population decoding classification accuracy for the pIT-matched typical vs atypical face-part configurations we used here was near chance level (50%) in pIT (*Figure 1C*, blue bar; by comparison, classification accuracy for face versus non-face objects classification was 99.6% using the same pIT sites). Further downstream in regions cIT and aIT, we found that the linear population decoding classification of these early pIT response-matched typical vs atypical face-part configurations was well above chance, suggesting that our pIT-matched face-part configuration detection challenge is largely solved somewhere between pIT and aIT (*Figure 1C*).

## Time course of responses in pIT for images with typical versus atypical arrangements of the face parts

We next closely examined the pIT neural response dynamics. To do this, we defined a face-part configuration preference value (d'; see Materials and methods) that measured each site's average selectivity for the typical face-part configurations relative to the atypical face-part configurations, and we asked how a given site's preference evolved over time (see alternative hypotheses in *Figure 1B*). First, we present three example sites which were chosen based on having the largest selectivity (absolute d') in the late phase (100–130 ms post image onset). In particular, most standard interpretations of face processing would predict a late phase preference for typical face-part configurations, if any preference were to develop (d' > 0). However, all three sites with the largest absolute d' preference had evolved a strong late phase preference for the atypical face-part configurations (d'<0) despite having had very similar, robust rising edge responses to both stimulus classes (response in early phase from 60 to 90 ms, shows that we were able to achieve relatively well matched stimuli from a feedforward perspective) (*Figure 2*, left column). Thus, these sites, which responded strongly overall to both stimulus classes derived from faces consistent with the overall face preference of sites in face-selective IT cortex (i.e. stronger responses to faces than to non-face objects; *Issa and DiCarlo, 2012*; *Tsao et al., 2006*), nonetheless demonstrated an additional (smaller) late modulation related to the configuration of the face parts. A late response modulation for atypical over typical face-part configurations was not restricted to the example sites as a majority of pIT sites (66%) responded more strongly to atypical over typical face-part configurations in the late response phase (prefer typical frontal face-part arrangement: 60–90 ms = 66% vs 100–130 ms = 34%; p = 0.000, n = 115) (*Figure 3B*, blue bars) even though almost all sites preferred faces over non-face objects throughout this time-course (60–90 ms = 98% vs 100–130 ms = 90%; p = 0.009, n = 115).

Next, we focused on the small but significant modulation of responses encoding the face-part configuration. Though this modulation can be relatively small compared to the absolute face response, the dynamics of face-part configuration selectivity (no change, increasing, decreasing) across the pIT population could provide insights into competing models of how additional, recurrent operations might modulate face processing (*Figure 1B*). In the adaptation and normalization models, we would expect no change in the average population selectivity, and the evidence accumulation, winner-take-all, or Bayesian inference models predict an increase in face configuration selectivity over the population over time. Instead, we found that many sites significantly decreased their typical face-part configuration preference over time similar to the three example sites. Of the 51 sites in our pIT sample that showed a significantly changing preference for typical face-part configurations over time (p < 0.01 criterion for significant change in d'), 84% of these sites showed a decreasing preference (n = 43 of 51 sites, p < $10^{-6}$, binomial test, n = 115 total sites) (*Figure 3A*, left column, light gray vs black lines). This surprising trend – decreasing relative response for typical face-part configurations versus atypical face-part configurations – was observed in both monkeys when analyzed separately ($p_{M1}$ = 0.000, $p_{M2}$ = 0.002, $n_{M1}$ = 43, $n_{M2}$ = 72 sites; *Figure 4A*).

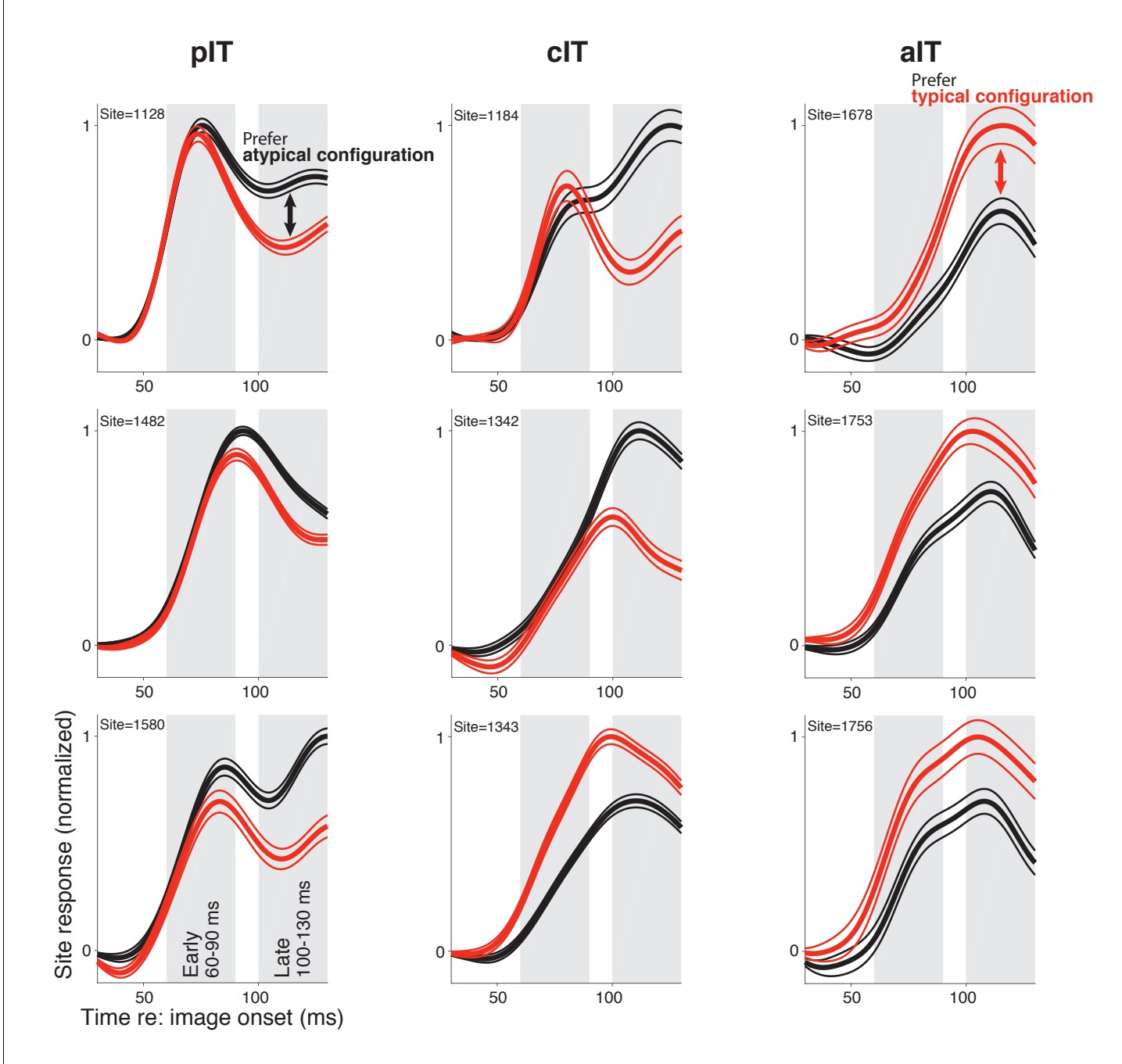

**Figure 2.** Responses in example sites to face-like images with typical and atypical face-part configurations. The three sites with the highest selectivity in the late response phase in each region are shown (pIT, cIT, and aIT; left, middle, and right columns, respectively) (d' selectivity measured in a 100–130 ms window, gray shaded region shown in bottom, left panel). While the three aIT sites (right column) demonstrated a late phase signal for the matched typical face context, the three pIT sites demonstrated the opposite preference in their late phase (100–130 ms) responses (red line = mean response of 8 images shown in *Figure 1B* red box, and black line = mean response of 13 images shown in *Figure 1B* black box).
DOI: https://doi.org/10.7554/eLife.42870.003

Specifically, sites still behaved like classic face-selective sites (responded more to faces than non-face objects) even in the late phase of the response (median d', 100–130 ms, faces vs non-face objects = 0.96 ± 0.06, n = 115 sites); however, we observed an additional, comparatively smaller response modulation encoding configuration information whereby typical face-part configurations

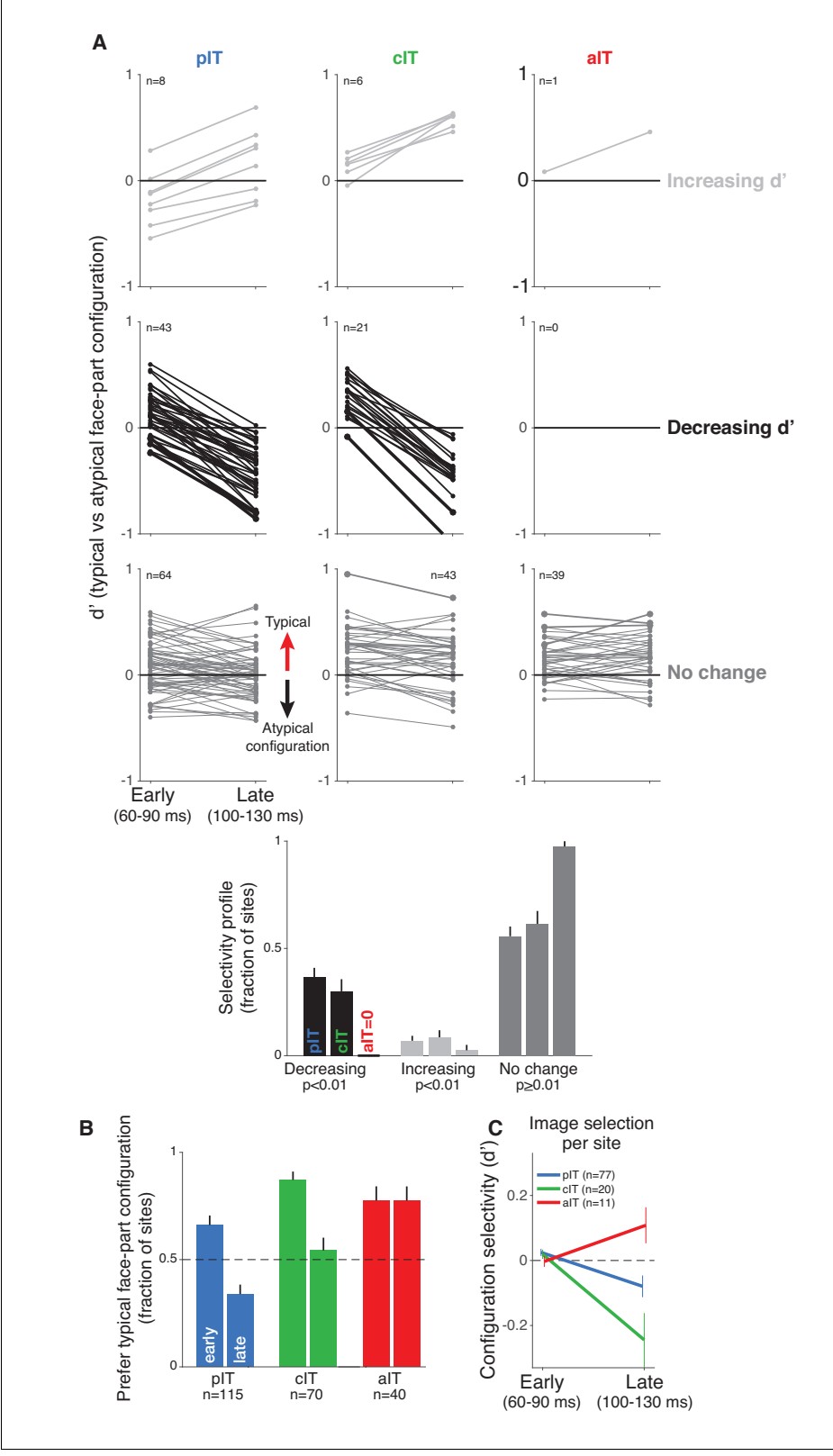

**Figure 3.** Time course of neural response preferences in pIT, cIT, and aIT for images with typical versus atypical face-part configurations. (**A**) Preferences for typical vs atypical part arrangements for each site are plotted in both early (60–90 ms post image onset) and late (100–130 ms) time windows. Sites are grouped based on region (pIT, cIT, aIT) and whether they showed a significant change in selectivity from early to late time windows (light gray = increased preference, black = decreased preference, and dark gray = no change in preference for typical versus atypical face-part

*Figure 3 continued on next page*

*Figure 3 continued*

configurations, significance tested at p < 0.01 level; example sites from *Figure 2* are plotted using thicker, darker lines). Many sites in pIT and cIT showed a decreasing signal for the typical face-part configuration context versus atypical configuration contexts over time (black lines, middle row, left and center panels). In contrast, no sites in aIT had this dynamic (middle row, right panel). (B) The fraction of sites whose responses showed a preference for images of typical, face-like arrangements of the face parts in pIT (blue), cIT (green) and aIT (red) in the early (60–90 ms) and late (100–130 ms) phase of the response. Note that, in the late phase of the response, most pIT neurons responded more strongly to atypical arrangements of face parts. (C) Selectivity measured for images driving similar responses within a site. This procedure ensured matched initial responses on a site-by-site basis rather than using a fixed set of images based on the overall population response (i.e. the fixed image set of *Figure 1B*; here, the initial d' for 60–90 ms is close to zero when images are selected site by site). Although initial response differences were near zero when using site based image selection, a late phase signal that was stronger for atypical face-part configurations still emerged in pIT and cIT but not in aIT similar to the decreasing selectivity profile observed when using a fixed image set for all sites.

DOI: https://doi.org/10.7554/eLife.42870.004

drove weaker responses relative to atypical face-part configurations across the population in the late response phase (median d', 100–130 ms, typical vs atypical face-part configurations = −0.12 ± 0.03, n = 115 sites). The observed trend for decreasing relative selectivity for typical face-part configurations over time over the population was driven by decreasing firing rates to the face images containing normally arranged face parts. Responses to these images were weaker by 18% on average in the late phase of the response compared to the early phase (Δrate (60–90 vs 100–130 ms) = −18% ± 4%, p = 0.000; n = 7 images) while responses to the images with atypical spatial arrangements of face parts – also capable of driving high early phase responses – did not experience any firing rate reduction in the late phase of the response (Δrate (60–90 vs 100–130 ms) = 2 ± 1%, p = 0.467; n = 13 images). The relative speed of this decreasing preference for typical face-part configurations

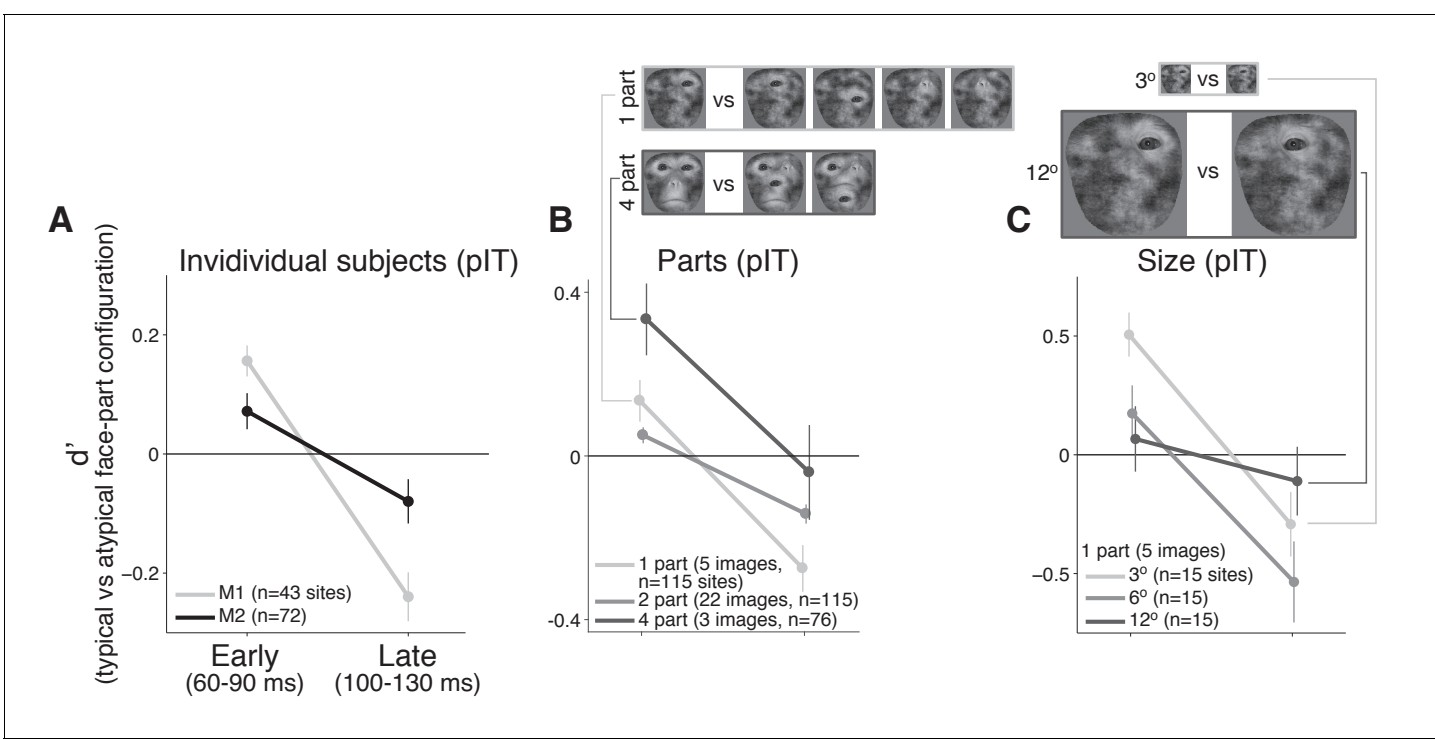

**Figure 4.** Individual monkey comparison and image controls for the decreasing selectivity profile in pIT. (A) Preference for images with the typical face-part configuration analyzed separately for each monkey. Median d' of pIT sites in both early and late time windows is shown. (B) Preference for images with typical versus atypical arrangements of the parts was re-computed using image subsets containing the same number of parts in the outline (the five 1-part and the three 4-part image subsets shown at top; the larger 2-part subset contained 30 images and is not shown). (C) The 1-part image subset was further tested at three different sizes (3°, 6°, and 12°). In all cases, pIT responses showed a decreasing preference over time for typically-arranged face parts leading to a preference for atypically arranged face parts in the later time window (100–130 ms).

DOI: https://doi.org/10.7554/eLife.42870.005

– starting 30 milliseconds after response onset – argues against arousal or attention mechanisms as those phenomena occur over long timescales, and our stimuli were randomized to avoid top-down priming effects for typical or atypical images.

The above observation of decreasing preference for typical face-part configurations over the pIT population seemed most consistent with predictions of error coding models (i.e. a conflict between the features represented locally and mismatched late-arriving predictions of those features from the face context), but one potential confound was that initial responses to typical and challenging, atypical configurations containing similar local features were not perfectly matched across the population (recall that we only required typical and atypical face-part configuration images to drive a response ≥90% of the whole face response). As a result, initial selectivity was non-zero (d'=0.11, n = 115 sites). This residual preference for the typical face-part configuration images may be small, but if this residual face selectivity is driven by nuisance dimensions, for example excess stimulus contrast in the typical face-part configuration class relative to the atypical face-part configuration class, then the typical configuration class may have experienced stronger activity dependent adaptation or normalization resulting in a decreasing typical face-part configuration preference over time. To more adequately limit general activity dependent mechanisms that could lead to decreasing responses to typical face-part configurations, we performed control analyses where initial activity was tightly matched per site or where the number of parts was matched across images.

## Controls in pIT for firing rate and low-level image variation

To strictly control for the possibility that simple initial firing rate differences could predict the observed phenomenon, we re-computed selectivity after first matching initial responses site-by-site. For this analysis, images were selected on a per site basis to evoke similar initial firing rates (images driving initial response within 25% of synthetic whole face response for that site, at least 5 images required per class). This image selection procedure virtually eliminated any differences in initial responses between the images of typical and atypical face-part configurations and hence any firing rate difference driven by potential differences in nuisance parameters when rearranging the face parts (*Figures 3C* and 60–90 ms), yet we still observed a significant drop in preference for images with typical frontal face-part arrangements versus atypical face-part arrangements in pIT ($\Delta_{d'}$ = −0.10 ± 0.03, p = 0.001, n = 77) (*Figure 3C*, blue line). Thus, the remaining dependence of firing rate dynamics on the image class and not on initial response strength argued against an exclusively activity based explanation to account for decreasing neural responses to typically configured face parts over time. Further arguing against this activity-dependent hypothesis, we found that the pattern of late phase population firing rates in pIT across images could not be significantly predicted from early phase pIT firing rates for each image ($\rho_{pIT\ early,\ pIT\ late}$ = 0.07 ± 0.17, p = 0.347; n = 20 images).

Thus far, we have performed analyses where images from the typical and atypical face-part configuration classes were similar in their initially evoked response which equated images at the level of neural activity but produced images varying in the number of parts. An alternative is to match the number of face parts between the typical and atypical configuration classes as another means of limiting the differences in nuisance dimensions such as the contrast, spatial frequency and retinal position of energy across images (see examples in *Figure 4B*). When we recomputed selectivity across subsets of images containing a matched number of one, two, or four parts (n = 5, 30, and 3 images, respectively), we still observed that pIT face selectivity decreased. For all three image subsets controlling the number of face parts, d' of the pIT population began positive on average in the sampled pIT population (i.e. preferred frontal face-part arrangements in 60–90 ms post-image onset) (median d' for 60–90 ms = 0.13 ± 0.05, 0.05 ± 0.02, 0.33 ± 0.09 for one, two, and four parts) and significantly decreased in the next phase of the response (100–130 ms post-image onset) becoming negative on average (median d' for 100–130 ms: −0.27 ± 0.06,−0.14 ± 0.02,−0.04 ± 0.12; one, two, four parts: p = 0.000, 0.000, 0.004, for d' comparisons between 60–90 ms and 100–130 ms, n = 115, 115, 76 sites) (*Figure 4B*). A similar decreasing face-part configuration selectivity profile was observed when we re-tested single part images at smaller (3°) and larger (12°) image sizes suggesting a dependence on the relative configuration of the parts and not on their absolute retinal location or absolute retinal size (median d' for 60–90 ms vs 100–130 ms: three degrees = 0.51 ± 0.09 vs −0.29 ± 0.14, twelve degrees = 0.07 ± 0.14 vs −0.11 ± 0.14; n = 15; p = 0.000, 0.025, 0.07) (*Figure 4C*). Thus, we suggest that the dynamic in pIT of a decreasing population selectivity for typical face-part configurations is a

fairly robust phenomenon specific to the face versus non-face configuration dimension as this dynamic persisted even when limiting potential variation across nuisance dimensions.

## Time course of responses in aIT and cIT for images with typical versus atypical arrangements of face parts

While previous studies have suggested the presence of putative error-like signals in the ventral visual cortex broadly agreeing with our present observations, none of these studies have recorded under the same experimental conditions using the same stimuli from areas that may provide the necessary prediction signals for computing the errors, leaving open the question of whether these signals are generated within the same area or could arrive from higher cortical areas (*Rao and Ballard, 1999*; *Schwiedrzik and Freiwald, 2017*). Here, we recorded from the anterior face-selective regions of IT which are furthest downstream of pIT and reflect additional stages of feedforward processing that could build selectivity for typical face-part configurations, a prerequisite for generating face predictions (see block diagram in *Figure 1B*). Indeed, in our aIT sample, the three sites with the greatest selectivity (absolute d') in the late response phase (100–130 ms) all displayed a preference for typical frontal face-part configurations (d' > 0) (*Figure 2*, right column). Also, in contrast to the dynamic selectivity profiles observed in many pIT sites, 98% of aIT sites (39 of 40) did not significantly change their relative preference for typical versus atypical configurations of the face parts from their initial feedforward response (p < 0.01 criterion for significant change at the site level) (*Figure 3A*, right column, bottom row, dark gray sites). Rather, we observed a stable selectivity profile over time in aIT (median d': 60–90 ms = 0.13 ± 0.03 vs 100–130 ms = 0.17 ± 0.03, p = 0.34, n = 40 sites). As a result, the majority of anterior sites preferred images with typical frontal configurations of the face parts in the late phase of the response (prefer typical face-part configuration: 60–90 ms = 78% of sites vs 100–130 ms = 78% of sites; p = 0.451, n = 40 sites; *Figure 3B*, red bars) despite only a minority (34%) of upstream sites in pIT preferring these images in their late response. Thus, spiking responses of individual aIT sites were as expected from a computational system whose purpose is to detect faces, as previously suggested (*Freiwald and Tsao, 2010*). Furthermore, the responses of aIT sites in this relatively early response window (60–130 ms post image onset) were too rapid and in the opposite direction (prefer typical face-part configurations) to be accounted for by late-arriving arousal or attention signals to the novel, atypical face-part configuration stimuli. In cIT whose anatomical location is intermediate to pIT and aIT, we observed many sites with decreasing selectivity (*Figure 2* and *3A*, middle columns), a dynamic that persisted even when we tightly matched initial responses on a site by site basis (*Figure 3C*, green line). The overall stimulus preference in cIT was intermediate to that of pIT and aIT (*Figure 3B*) consistent with the intermediate position of cIT in the IT hierarchy.

To further test whether downstream areas cIT and aIT could be candidates for the putative prediction signals underlying face part prediction errors in pIT, we examined whether early response patterns in cIT and aIT were correlated to the later response in pIT. Interestingly, we found that the turning profiles across images in the early response phases of cIT and aIT were significant predictors of late phase activity in pIT ($\rho_{cIT\ early,\ pIT\ late}$ = -0.52 ± 0.11, p = 0.000; $\rho_{aIT\ early,\ pIT\ late}$ = -0.36 ± 0.14, p = 0.012; $n_{pIT}$ = 115, $n_{cIT}$ = 70, $n_{aIT}$ = 40 sites; n = 20 images), even better predictors than early phase activity in pIT itself ($\rho_{pIT\ early,\ pIT\ late}$ = 0.07 ± 0.17, p = 0.347). That is, for images that produced high early phase responses in cIT and aIT, the following later phase responses of units in the lower level area (pIT) tended to be low, consistent with error coding models which posit that feedback from higher areas (in the form of predictions of the face features) would contribute to the decreasing activity observed in lower areas encoding those face features.

## Computational models of neural dynamics in IT

We next proceeded to formalize the conceptual ideas introduced in *Figure 1B* and build neurally mechanistic, dynamical models of gradually increasing complexity to determine the minimal set of assumptions that could capture our empirical findings of non-trivial, dynamic selectivity changes during face detection across face-selective subregions in IT. This modeling effort is only intended to present at least one formal, working model of the observed population dynamics in IT which could complement previously reported phenomena in the literature that lacked a quantitative modeling framework. We submit that our model will inherently be underconstrained given the present, limited data. Much further circuit dissection work would need to be done to identify the sources of dynamics

in pIT as these could arrive from downstream areas (as suggested by our correlative data between cIT/aIT early responses and pIT late responses) or could be shaped by lateral recurrences in pIT or both. Nonetheless, we leverage normative modeling principles for feedforward hierarchal processing and top-down hierarchical prediction (e.g. predictive coding, hierarchical Bayesian inference) to define at least one model class that can account for our data.

Previous functional and anatomical data show that the face-selective subregions in IT are connected forming an anterior to posterior hierarchy and show that pIT serves as the primary input into this hierarchy (*Moeller et al., 2008*; *Freiwald and Tsao, 2010*; *Grimaldi et al., 2016*). Thus, we evaluated dynamics in different hierarchical architectures using a linear dynamical systems modeling framework where pIT, cIT, and aIT act as sequential stages of processing (network diagrams in *Figure 5* and see Materials and methods). A core principle of feedforward ventral stream models is that object selectivity is built by stage-wise feature integration in a manner that leads to relatively low dimensional representations at the top of the hierarchy abstracted from the high-dimensional input layer. We were interested in how signals temporally evolve across a similar architectural layout. We used the simplest feature integration architecture where a unit in a downstream area linearly sums the input from units in an upstream area, and we stacked this computation to form three layer networks (*Figure 5*). This simple, generic feedforward encoding model conceptualizes the idea that different types of evidence, local and global (i.e. information about the parts and the relative spatial arrangement of parts), have to converge and be integrated to separate typical from atypical face-part configurations in our image set. We used linear networks as monotonic nonlinearities can be readily accommodated in our framework (*Seung, 1997*; *Rao and Ballard, 1999*; also see *Figure 6*). Importantly, we used a simple encoding scheme as our goal was not to build full-scale deep neural network encoding models of image representation (*Yamins et al., 2014*) but to bring focus to an important biological property that is often not considered in deep nets, neural dynamics.

We implemented a range of ideas previously proposed in the literature. The functional implications of these ideas were highlighted in *Figure 1B*; at a mechanistic level, these functional properties can be directly realized via different recurrent processing motifs between neurons (*Figure 5B*, base feedforward model was augmented with recurrent connections to form new models). We focus on error encoding models since the observed neural phenomena in pIT and their relationship to responses in cIT and aIT suggested the generation of a prediction error of pIT preferred local features in late phase pIT responses. Here, we asked whether a dynamical systems implementation of error coding in a hierarchical prediction network could account for the temporal response patterns observed neurally. To constrain our choice of an error coding model, we took a normative approach minimizing a quadratic reconstruction cost between stages (top stages predict their input stages) as the classical reconstruction cost is at the core of an array of hierarchical generative models including hierarchical Bayesian inference (*Lee and Mumford, 2003*), Boltzmann machines (*Ackley et al., 1985*), analysis-by-synthesis networks (*Seung, 1997*), sparse coding (*Olshausen and Field, 1996*), predictive coding (*Rao and Ballard, 1999*), and autoencoders in general (*Rifai et al., 2011*). Optimizing a quadratic loss results in feedforward and feedback connections that are symmetric – reducing the number of free parameters – such that inference on the represented variables at any intermediate stage is influenced by both bottom-up sensory evidence and current top-down interpretations. Critically, a common feature of this large model family is the computation of between-stage error signals via feedback, which is distinct from state-estimating model classes (i.e. feedforward models) that do not compute or propagate errors. A dynamical implementation of such a network uses leaky integration of error signals which, as shared computational intermediates, guide gradient descent of the values of the represented variables to a previously learned target value (Δactivity of each neuron => online inference) or descend the connection weights to values that give the best future behavior (Δsynaptic strengths => offline learning), here defined as an unsupervised reconstruction goal (similar results were found using other goals and networks such as supervised discriminative networks; see *Figure 6*).

We found that the dynamics of error signals in our hierarchical model naturally displayed a strong decrease of selectivity in a sub-component of its first processing stage – qualitatively similar behavior to the selectivity decrease that we observed in many pIT and cIT neural sites (*Figure 5A*, second column, blue and green curves). These error signals integrate converging state signals from two stages – one above (prediction) and one below (sensory evidence). The term 'error' is thus meaningful in the hidden processing stages where state signals from two stages can converge. The top nodes of a

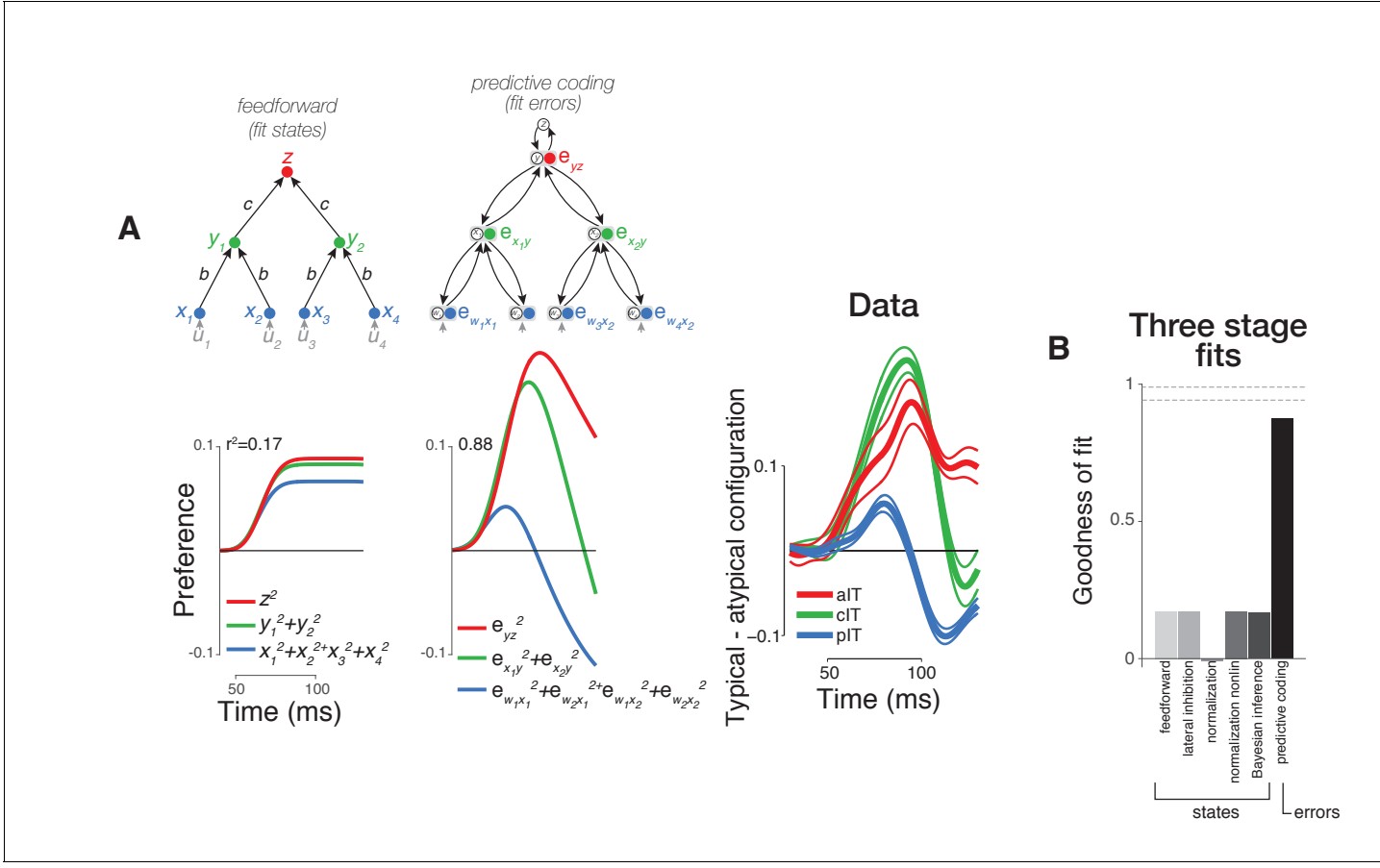

**Figure 5.** Computational modeling of neural dynamics in IT. (**A**) Three stage neural networks with recurrent dynamics were constructed to model neural signals measured in pIT, cIT, and aIT corresponding to the first (blue), second (green), and third (red) model processing stages (top row; see Materials and methods). Models received four inputs (gray) into four hidden stage units (blue) which sent feedforward projections that converged onto two units in the next layer (green) (self-connections reflecting leak currents are not shown here for clarity). State (feature) coding models generally showed increasing selectivity over time from hidden to output layers as exemplified by the feedforward model (left) and did not demonstrate the strong decrease of stimulus preference in their hidden processing stage as observed in the pIT and cIT neural population (blue and green lines, feedforward model shown). However, the neurons coding errors in a feedback-based hierarchical model did show a strong decrease of stimulus preference in the hidden processing stage (second column; reconstruction errors instead of the states were fit directly to the data). This model which codes the error signals (filled circles) also codes the states (open circles). Far right, population averaged neural selectivity profile for difference between typical, frontal versus atypical face-part arrangements (normalized by the mean population response to the whole face) used in model fitting (best fitting feedforward and error coding models are shown at left). (**B**) Goodness of fit of all three stage models tested to population averaged selectivity profiles (dashed lines represent mean and standard error of reliability of neural data as estimated by bootstrap resampling). Besides the base feedforward architecture, additional excitatory feedback (Bayesian inference) or lateral inhibitory (lateral inhibition or normalization) connections between units were implemented to produce recurrent dynamics. The goodness of fit to the population averaged neural data (far right in (**A**)) of the state coding models (first five bars) and of the reconstruction error coding model (last bar) are shown.

DOI: https://doi.org/10.7554/eLife.42870.006

hierarchy receive little descending input and hence do not carry additional errors with respect to the desired computation; rather, top nodes convey the face predictions that influence errors in bottom nodes. This behavior in the higher processing stages is consistent with our observation of explicit representation of faces in aIT in all phases of the response (*Figures 2–3*, aIT data) and with similar observations of decodable identity signals by others in all phases of aIT responses for faces (*Meyers et al., 2015*) and objects (*Hung et al., 2005*; *Majaj et al., 2015*). We also found similar error dynamics when using a simpler two-layer network as opposed to three layers suggesting that these error signal dynamics along with prediction signals emerge even in the simplest cascaded architecture (*Figure 6*).

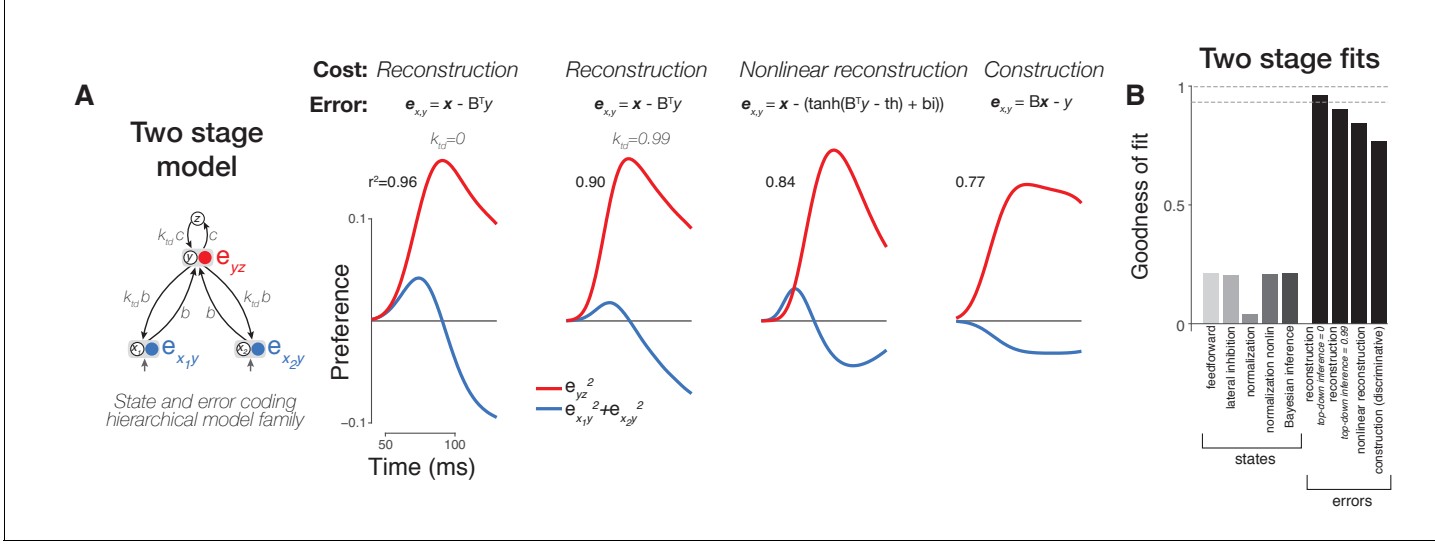

**Figure 6.** Comparison of variants of error coding hierarchical models that use different algorithms for online inference. (**A**) Additional varieties of error computing networks can be generated by varying the online inference algorithm that they use. In one case, inference does not utilize top-down information between stages (classic error backpropagation; between-stage feedback connections shown are not used in these networks during runtime). On the other hand, between-stage feedback can be used to optimize online estimates such as in more general forms of error backpropagation and predictive coding. We approximated these two extremes by including a parameter ($k_{td}$, see Materials and methods) controlling the relative weighting of bottom-up (feedforward) and top-down (feedback) evidence during online inference (first and second panels). We found that top-down inference between stages was not necessary to produce the appropriate error signal dynamics, and $k_{td}$ was equal to zero (similar to the lack of inference in classic error backpropagation) in our best fitting two-layer (first column) and three-layer (*Figure 5A*, second column) models although models with $k_{td} \sim 1$ also performed well (second column). Models can also differ in their goal (cost function) which directly impacts the error signals computed (equations in top row). Under a nonlinear reconstruction goal (emulating the nonlinear nature of spiking output), the resulting error signals were still consistent with our data (third column). A simple sigmoidal nonlinearity, however, did lead to additional details present in our neural data such as a rapid return of stimulus preference to zero in the hidden layer. When we tested a discriminative, construction goal more consistent with a supervised learning setting where high-dimensional bottom-up responses simply have to match a low-dimensional downstream target signal as in classification tasks, we found that the errors of construction did not match the data as well as reconstruction errors (compare fourth column to first three columns). (**B**) Goodness of fit to population neural data for all two stage models including two layer versions of state (feature) coding controls (same format as *Figure 5B*).

DOI: https://doi.org/10.7554/eLife.42870.007

For control comparisons, we also implemented a range of feature coding models beginning with a basic feedforward model and augmenting it with lateral connections (winner-take-all lateral inhibition and normalization models) or feedback connections (hierarchical Bayesian inference) (*Carandini et al., 1997*; *Seung, 1997*). However, all of these state coding control models failed to reproduce the observed neural dynamics across the ventral visual hierarchy. Rather, the selectivity of these models simply increased to a saturation level set by the leak term (shunting inhibition) in the system as in the strictly feedforward model (*Figure 5A*, first column). That adding normalization proved insufficient to generate the observed neural dynamics can be explained by the fact that the normalized response to a stimulus cannot easily fall below the response to a stimulus that was initially similar in strength. Thus, a decreasing average preference for a stimulus across a population of cells (i.e. *Figures 2–4*, pIT data) for similar levels of average input is difficult when only using a basic normalization model mediated by surround (within-stage) suppression.

## Discussion

We have measured neural responses during a difficult discrimination between images with typical and atypical face-part configurations across the IT hierarchy and demonstrated that the population preference for normally configured face parts in the intermediate (a.k.a hidden) processing stages decreases over time – that is population responses at lower levels of the hierarchy (pIT and cIT) signal deviations of their preferred features from their expected configuration whereas the top level

(aIT) rapidly developed and then maintained a preference for natural, frontal face-part configurations. The relative speed of selectivity changes in pIT makes high-level explanations based on fixational eye movements or shifts in attention (e.g. from behavioral surprise to unnatural arrangements of face parts) unlikely as saccades and attention shifts occur on slower timescales (hundreds of milliseconds) (*Egeth and Yantis, 1997*; *Müller et al., 1998*; *Ward et al., 1996*) than the ~30 ms dynamical phenomena we observed. The presence of stronger responses to typical than to atypical face-part configuration images in aIT further argues against general arousal effects which would have predicted stronger not weaker responses to surprising, atypical images in aIT. Rather, the rapid propagation of neural signals over tens of milliseconds suggested intracortical processing within the ventral visual stream in a manner that was not entirely consistent with a pure feedforward model, even when we included strong nonlinearities in these models such as normalization and even when we stacked these operations to form more complex three stage models. However, augmenting the feedforward model so that it represented the prediction errors generated during hierarchical processing of atypical configurations produced the observed neural dynamics and hierarchical pattern of signal propagation (*Figures 5–6*). This view argues that many IT neurons code error signals. However, the exact mechanism for producing prediction errors remains to be determined. While we showed that a recurrent model could recapitulate the observed signals, how this model maps to the IT network is unclear. Recurrence could be implemented by a circuit within pIT which computes the predictions that lead to prediction errors within the same region. Whether the error computation is done internally in pIT or depends on downstream sources such as cIT or aIT can be directly tested by causal knock-outs of cIT or aIT.

## Comparison to previous neurophysiology studies in IT

Multiple visual neurophysiology studies have shown evidence of neural responses consistent with error signals. This includes the seminal predictive coding study on end-stopping in V1 (*Rao and Ballard, 1999*). More recently, studies in IT have used pairing of images over time to create sequences with predictable temporal structure and found evidence of putative error signals when those statistically exposed temporal predictions were violated (*Meyer et al., 2014*; *Schwiedrzik and Freiwald, 2017*). Our work expands those IT findings in three directions. First, we revealed error coding dynamics that are naturally present in the system without using statistical exposure or behavioral training to induce signals. Second, errors depended on the spatial statistics of the features – rather than depending on temporal statistics – which may be more directly related to native, spatial form processing in IT. Third and perhaps most importantly, we identified a putative source of face prediction signals in downstream IT by recording from multiple areas in the same experiment and showing that early signals in anterior areas were (negatively) correlated with late signals in pIT. Together, these advances suggest a more definitive role of error signaling in natural, online vision. Formalizing this claim, we found that the pattern of observed dynamics in pIT, cIT, and aIT were indeed difficult to account for quantitatively when using feature coding models but could be computationally modeled at a population level using a simple error coding model (*Figure 5*). Extensions of our dynamical modeling framework to more realistic large-scale networks could be useful for future studies of IT response dynamics.

Our suggestion that many IT neurons code errors is consistent with the observation of strong responses to extremes in face space (*Leopold et al., 2006*) providing an alternative interpretation to the prior suggestion that cIT neurons are not tuned for typical faces but are instead tuned for atypical face features (i.e. extreme feature tuning) (*Freiwald et al., 2009*). In that prior work, the response preference of each neuron was determined by averaging over a long time window (~200 ms). By looking more closely at the fine time scale dynamics of the IT response, we suggest that this same extreme coding phenomenon can instead be interpreted as a natural consequence of networks that have an actual tuning preference for face features in typical configurations (as evidenced by an initial, feedforward response preference for typical frontal faces in pIT, cIT, and aIT; *Figure 3B*) but that also compute error signals with respect to that preference. Under the present hypothesis, some IT neurons are preferentially tuned to typical spatial arrangements of face features, and other IT neurons are involved in coding errors with respect to those typical arrangements. We speculate that these intermixed state estimating and error coding neuron populations are both sampled in standard neural recordings of IT, even though only state estimating neurons are truly reflective of the tuning preferences of that IT processing stage. The precise fractional contribution of errors to total

neural activity is difficult to estimate from our data. Under the primary image condition tested, not all sites significantly decreased their selectivity (~60% did not change their selectivity). We currently interpret these sites as coding state (feature) estimates (*Figure 3A*, light and dark gray lines in top and bottom rows, respectively). Alternatively, at least some of the non-reversing sites might be found to code errors under other image conditions than the one that we tested. Furthermore, in our primary image condition, selectivity decreases only accounted for ~15% of the overall spiking. However, at a computational level, the absolute contribution of error signals to spiking may not be the critical factor as even a small relative contribution may have important consequences in the network.

## Comparison across dynamical models of neural processing

Our goal was to test a range of existing recurrent models by recording neural dynamics across multiple cortical stages which provided stronger constraints on computational models than fitting neural responses from only one area as in prior work (*Carandini et al., 1997*; *Rao and Ballard, 1999*). Crucially, we found that the multi-stage neural dynamics observed in our data could not be adequately fit by only using lateral recurrences such as adaptation, lateral inhibition, and standard forms of normalization (*Figure 5B*). These results did not change when we made our simple networks more complex by adding more stages (compare *Figure 5* versus *Figure 6*) or by using more realistic model units with monotonic nonlinearities similar to a spiking nonlinearity (data not shown). Indeed, we specifically chose our stimuli to evoke similar levels of within stage neural activity to limit the effects of known mechanisms that depend on activity levels through lateral interactions (e.g. adaptation, normalization), and we fully expect that these activity dependent mechanisms would operate in parallel to top-down, recurrent processes during general visual processing. We emphasize that we only tested the standard form of normalization as originally proposed, using within stage pooling and divisive mechanisms (*Carandini et al., 1997*). Since that original mechanistic formulation, normalization has evolved to become a term that broadly encapsulates many forms of suppression phenomena and can include both lateral interactions within an area and feedback interactions from other areas (*Nassi et al., 2014*; *Coen-Cagli et al., 2015*). Thus, while our results do not follow from the original mechanistic form of normalization, they may yet fall under normalization more broadly construed as a term for suppression phenomena (error coding would require a similar suppressive component). Here, we have provided a normative model for how top-down suppression would follow from the well-defined computational goals of many hierarchical neural network models. Finally, we clarify that any top-down interactions instantiated in coding errors need not originate in other areas but could happen within the same area (e.g. layer 2/3 predictions interacting with layer 4) which could be viewed as a local feedback interaction with respect to the whole network, and this is a testable mechanistic hypothesis that is not ruled out by the present work.

## Computational utility of coding errors in addition to states

The present study provides evidence that errors are not only computed, but that they might be explicitly encoded in spiking rates. We emphasize that this result at the level of population neural dynamics was robust across choices of cost function; we tested models with different unsupervised and supervised performance errors (reconstruction, nonlinear reconstruction, and discriminative) and found similar population level error signals across these networks (*Figure 6*). Thus, errors as generally instantiated in the state-error coding hierarchical model family provide a good approximation to IT population neural dynamics. In error-computing networks, errors provide control signals for guiding learning giving these networks additional adaptive power over basic feature estimation networks. This property helps augment the classical, feature coding view of neurons which, with only feature activations and Hebbian operations, does not lead to efficient learning in the manner produced by gradient descent using error backpropagation (*Rumelhart et al., 1986*). Observation of error signals may provide insight into how more intelligent unsupervised and supervised learning algorithms such as backpropagation could be plausibly implemented in the brain. A potentially important contribution of this work is the suggestion that gradient descent algorithms are facilitated by using an error code so that efficient learning is reduced to a simple Hebbian operation at synapses and efficient inference is simply integration of inputs at the cell body (see *Equation 10* and text in Materials and methods). This representational choice, to code the computational primitives of

gradient descent in spiking activity, would simply leverage the existing biophysical machinery of neurons for inference and learning.

## Materials and methods

### Animals and surgery

All surgery, behavioral training, imaging, and neurophysiological techniques are identical to those described in detail in previous work (*Issa and DiCarlo, 2012*). Two rhesus macaque monkeys (*Macaca mulatta*) weighing 6 kg (Monkey 1, female) and 7 kg (Monkey 2, male) were used. A surgery using sterile technique was performed to implant a plastic fMRI compatible headpost prior to behavioral training and scanning. Following scanning, a second surgery was performed to implant a plastic chamber positioned to allow targeting of physiological recordings to posterior, middle, and anterior face patches in both animals. All procedures were performed in compliance with National Institutes of Health guidelines and the standards of the MIT Committee on Animal Care and the American Physiological Society.

### Behavioral training and image presentation

Subjects were trained to fixate a central white fixation dot during serial visual presentation of images at a natural saccade-driven rate (one image every 200 ms). Although a 4° fixation window was enforced, subjects generally fixated a much smaller region of the image (<1°) (*Issa and DiCarlo, 2012*). Images were presented at a size of 6° except for control tests at 3° and 12° sizes (*Figure 4C*), and all images were presented for 100 ms duration with 100 ms gap (background gray screen) between each image. Images were presented in a randomly interleaved fashion at this rate of 5 images per second, so subjects could not predict the image class (e.g. face vs non-face or typical vs atypical face-part configuration) and were more likely to engage automatic processing of the visual stimuli. Up to 15 images were presented during a single fixation trial, and the first image presentation in each trial was discarded from later analyses. Five repetitions of each image in the general screen set were presented, and ten repetitions of each image were collected for all other image sets. The screen set consisted of a total of 40 images drawn from four categories (faces, bodies, objects, and places; 10 exemplars each) which was used to derive a measure of face versus non-face object selectivity (faces versus bodies, objects, and places grouped together).

Following the screen set testing, some sites were tested using an image set containing images of face parts presented in different combinations and positions (*Figure 1B*, left panel). We first segmented the face parts (eye, nose, mouth) from a monkey face image. These parts were then blended using a Gaussian window, and the face outline was filled with pink noise to create a continuous background texture. A face part could appear on the outline at any one of nine positions on an evenly spaced $3 \times 3$ grid. Although the number of possible images is large ($4^9$ = 262,144 images), we chose a subset of these images for testing neural sites (n = 82 images). Specifically, we tested the following images: the original whole face image, the noise-filled outline, the whole face reconstructed by blending the four face parts with the outline, all possible single part images where the eye, nose, or mouth could be at one of nine positions on the outline (n = $3 \times 9$ = 27 images), all two part images containing a nose, mouth, left eye, or right eye at the correct outline-centered position and an eye tested at all remaining positions (n = 4*8–1 = 31 images), all two part images containing a correctly positioned contralateral eye while placing the nose or mouth at all other positions (n = 2*8–2 = 14 images), and all correctly configured faces but with one or two parts missing besides those already counted above (n = 4 + 3 = 7 images). The particular two-part combinations tested were motivated by prior work demonstrating the importance of the eye in early face processing (*Issa and DiCarlo, 2012*), and we sought to determine how the position of the eye relative to the outline and other face parts was encoded in neural responses. The three and four part combinations were designed to manipulate the presence or absence of a face part for testing the integration of face parts, and in these images, we did not vary the positions of the parts from those in a naturally occurring face. In a follow-up test on a subset of sites, we permuted the position of the four face parts under the constraint that they still formed the configuration of a naturally occurring face (i.e. preserve the 'T' configuration, n = 10 images; *Figure 4B*). We tested single part images at 3° and 12° sizes in a subset of sites (n = 27 images at each size; *Figure 4C*).

## MR imaging and neurophysiological recordings

Both structural and functional MRI scans were collected in each monkey. Putative face patches were identified in fMRI maps of face versus non-face object selectivity in each subject. A stereo microfocal x-ray system (*Cox et al., 2008*) was used to guide electrode penetrations in and around the fMRI defined face-selective subregions of IT. X-ray based electrode localization was critical for making laminar assignments since electrode penetrations are often not perpendicular to the cortical lamina when taking a dorsal-ventral approach to IT face patches. Laminar assignments of recordings were made by co-registering x-ray determined electrode coordinates to MRI where the pial-to-gray matter border and the gray-to-white matter border were defined. Based on our prior work estimating sources of error (e.g. error from electrode tip localization and brain movement), registration of electrode tip locations to MRI brain volumes has a total of <400 micron error which is sufficient to distinguish deep from superficial layers (*Issa et al., 2013*). Multi-unit activity (MUA) was systematically recorded at 300 micron intervals starting from penetration of the superior temporal sulcus such that all sites at these regular intervals were tested with a screen set containing both faces and non-face objects, and a subset of sites that were visually driven were further tested with our main image set manipulating the position of face parts. Although we did not record single-unit activity, our previous work showed similar responses between single-units and multi-units on images of the type presented here (*Issa and DiCarlo, 2012*), and our results are consistent with observations in previous single-unit work in IT (*Freiwald et al., 2009*). Recordings were made from PL, ML, and AM in the left hemisphere of monkeys 1 and 2 and additionally from AL in monkey 2. AM and AL are pooled together in our analyses forming the aIT sample while PL and ML correspond to the pIT and cIT samples, respectively.

## Neural data analysis

The face patches were physiologically defined in the same manner as in our previous study (*Issa and DiCarlo, 2012*). Briefly, we fit a graded 3D sphere model (linear profile of selectivity that rises from a baseline value toward the maximum at the center of the sphere) to the spatial profile of face versus non-face object selectivity across our sites. We tested spherical regions with radii from 1.5 to 10 mm and center positions within a 5 mm radius of the fMRI-based centers of the face patches. The resulting physiologically defined regions were 1.5 to 3 mm in diameter. Sites which passed a visual response screen (mean response in a 60–160 ms window >2*SEM above baseline for at least one of the four categories in the screen set) were included in further analysis. All firing rates were baseline subtracted using the activity in a 25–50 ms window following image onset averaged across all repetitions of an image. Finally, given that the visual response latencies in monkey two were on average 13 ms slower than those in monkey one for corresponding face-selective regions, we applied a single latency correction (13 ms shift to align monkey 1 and monkey 2's data) prior to averaging across monkeys. This was done so as not to wash out any fine timescale dynamics by averaging. Similar results were obtained without using this latency correction as dynamics occurred at longer timescales (~30 ms). This single absolute adjustment was more straightforward than the site-by-site adjustment used in our previous work (*Issa and DiCarlo, 2012*) (though similar results were obtained using this alternative latency correction). Even when each monkey was analyzed separately, we still observed pIT selectivity dynamics (*Figure 4A*). Furthermore, there was <10 ms average latency difference between pIT, cIT, and aIT so that a common 30 ms wide analysis window for early (60–90 ms) and late (100–130 ms) firing rates was sufficient across IT stages. Images that produced an average population response $\geq 0.9$ of the initial response (60–100 ms) to a face image with all face parts arranged in their typical positions in a frontal face were analyzed further (*Figures 2* and *3*). Stimulus selection was intended to limit potentially confounding differences in visual drive between image classes. In a control test, we also repeated our analysis by selecting images on a site-by-site basis where images with typical frontal and atypical arrangements of face parts were chosen to be within 0.75x to 1.25x of the initial response to the complete face image (minimum of five typical and five atypical images in this response range for inclusion of site in analysis). In follow-up analyses of population responses, we specifically limited comparison to images with the same number of parts (*Figure 4B,C*). For example, for single part images, we used the image with the eye in the upper, contralateral region of the outline as a reference requiring a response $\geq 0.9$ of the initial population response to this reference for inclusion of the images in this analysis. We found that four other

images of the 27 single-part images elicited a response at least as large as 90% of the response to this standard image. For images containing all four face parts, we used the complete, frontal face as the standard and found atypical face-part arrangements of the four face parts that drove at least 90% of the early response to the whole face (2 images out of 10 tested). To measure decoding performance for typical versus atypical face-part configurations (or face versus non-face objects from our screen set), we used a linear-SVM classifier trained on responses (60–200 ms post image onset) of resampled subsets of 30 sites from pIT, cIT, or aIT. Trials splits were used so that all images were used in training and tested but on separate, held-out trials (90% train, 10% test). To compute individual site d′ for each stimulus partition (e.g. typical versus atypical arrangements of 1 face part), we combined all presentations of images with frontal face-part arrangements and compared these responses to responses from all presentations of images with atypical face-part arrangements using $d' = (u_1 - u_2)/((var_1 + var_2)/2)^{1/2}$ where variance was computed across all trials for that image class (e.g. all presentations of all typical face-part configuration images); this was identical to the d′ measure used for face versus non-face object selectivity in *Figure 1* and Results and to that used in previous work for computing selectivity for faces versus non-face objects (*Aparicio et al., 2016*; *Ohayon et al., 2012*). For example, for the main image set (*Figure 1B*), we compared all presentations of typical face-part configuations (8 images x 10 presentations/image = 80 total presentations) to all presentations of atypical face-part arrangements (13 images x 10 presentations/image = 130 total presentations) to compute the d′ values for each site in two time windows (60–90 ms and 100–130 ms) as shown in *Figure 3A*. A positive d′ implies a stronger response to more naturally occurring typical frontal arrangements of face parts while a negative d′ indicates a preference for atypical arrangements of the face parts.

## Dynamical models

### Modeling framework and equations

To model the dynamics of neural response rates in a hierarchy, we start with the simplest possible model that might capture those dynamics: a model architecture consisting of a hidden stage of processing containing two units that linearly converge onto a single output unit. We use this two-stage cascade for illustration of the basic concepts which can be easily extended to longer cascades with additional stages, and we ultimately used a three-stage version of the model to fit our neural data collected from three cortical stages (*Figure 5*).

An external input is applied separately to each hidden stage unit, which can be viewed as representing different features for downstream integration. We vary the connections between the two hidden units within the hidden processing stage (lateral connections) or between hidden and output stage units (feedforward and feedback connections) to instantiate different model families. The details of the different architectures specified by each model class can be visualized by their equivalent neural network diagrams (*Figure 5*). Here, we provide a basic description for each model tested using the two stage example for simplicity. All two stage models utilized a 2 × 2 feedforward identity matrix $A$ that simply transfers inputs $u$ (2 × 1) to hidden layer units $x$ (2 × 1) and a 1 × 2 feedforward vector $B$ that integrates hidden layer activations $x$ into a single output unit $y$.

$$A = aI, B = b[1,1] \tag{1}$$

By simply substituting in the appropriate unit vector and weight matrix transforming inputs from one layer to the next for the desired network architecture, this simple two-stage architecture can be extended to larger networks (e.g. see three-stage network diagrams in *Figure 5A*). To generate dynamics in the simple networks below, we assumed that neurons act as leaky integrators of their total synaptic input, a standard rate-based model of a neuron used in previous work (*Seung, 1997*), (*Rao and Ballard, 1999*).

### Pure feedforward

In the purely feedforward family, connections are exclusively from hidden to output stages through feedforward matrices $A$ and $B$.

$$\dot{\mathbf{x}} = A\mathbf{u} - \mathbf{x}/\tau, \ \dot{y} = B\mathbf{x} - y\tau \tag{2}$$

where τ is the time constant of the leak current which can be seen as reflecting the biophysical limitations of neurons (a perfect integrator with large τ would have almost no leak and hence infinite memory).

## Lateral inhibition
Lateral connections (matrix with off-diagonal terms) are included and are inhibitory. The scalar $k_l$ sets the relative strength of lateral inhibition versus bottom-up input.

$$\dot{\mathbf{x}} = A\mathbf{u} - \begin{bmatrix} 0 & k_1 \\ k_1 & 0 \end{bmatrix}\mathbf{x} - \mathbf{x}/\tau, \; \dot{y} = B\mathbf{x} - y/\tau \tag{3}$$

## Normalization
An inhibitory term that scales with the summed activity of units within a stage is included. The scalar $k_s$ sets the relative strength of normalization versus bottom-up input.

$$\dot{\mathbf{x}} = A\mathbf{u} - k_s \sum x \cdot \mathbf{x}/\tau - \mathbf{x}/\tau, \; \dot{y} = B\mathbf{x} - k_s y \cdot y - y/\tau \tag{4}$$

## Normalization (nonlinear) (*Carandini et al., 1997*)
The summed activity of units within a stage is used to nonlinearly scale shunting inhibition.

$$\dot{\mathbf{x}} = A\mathbf{u} - \frac{x}{\tau\sqrt{1 - k_s \sum x}}, \; \dot{y} = B\mathbf{x} - \frac{y}{\tau\sqrt{1 - k_s y}} \tag{5}$$

Note that this is technically a nonlinear dynamical system, and since the normalization term in *Equation (5)* is not continuously differentiable, we used the fourth-order Taylor approximation around zero in the simulations of *Equation (5)*.

## Feedback (linear reconstruction)
The feedback-based model is derived using a normative framework that performs optimal inference in the linear case (*Seung, 1997*) (unlike the networks in *Equations (2)-(5)* which are motivated from a mechanistic perspective but do not directly optimize a squared error performance loss). The feedback network minimizes the cost $C$ of reconstructing the inputs of each stage (i.e. mean squared error of layer *n* predicting layer *n-1*).

$$C = \frac{1}{2}(\mathbf{u} - A^T\mathbf{x})^2 + \frac{1}{2}(\mathbf{x} - B^T\mathbf{y})^2 \tag{6}$$

Differentiating this coding cost with respect to the encoding variables in each layer *x*, *y* yields:

$$\frac{\partial C}{\partial x} = A(\mathbf{u} - A^T\mathbf{x}) + (\mathbf{x} - B^T\mathbf{y}), \; \frac{\partial C}{\partial \mathbf{y}} = -B(\mathbf{x} - B^T\mathbf{y}) \tag{7}$$

The cost function $C$ can be minimized by descending these gradients over time to optimize the values of *x* and *y*:

$$\begin{aligned} \frac{dx}{dt} &= -\frac{\partial C}{\partial x} = A(\mathbf{u} - A^T\mathbf{x}) - (\mathbf{x} - B^T\mathbf{y}) - \mathbf{x}/\tau \\ \frac{dy}{dt} &= -\frac{\partial C}{\partial y} = B(\mathbf{x} - B^T\mathbf{y}) - \mathbf{y}/\tau \end{aligned} \tag{8}$$

The above dynamical equations are equivalent to a linear network with a connection matrix containing symmetric feedforward ($B$) and feedback ($B^T$) weights between stages *x* and *y* as well as within-stage pooling followed by recurrent inhibition ($-AA^T x$ and $-BB^T y$) that resembles normalization. The property that symmetric connections minimize the cost function $C$ generalizes to a feedforward network of any size or number of hidden processing stages (i.e. holds for arbitrary lower triangular network connection matrices). The final activation states (*x,y*) of the hierarchical generative network are optimal in the sense that the bottom-up activations (implemented through feedforward connections) are balanced by the top-down expectations (implemented by feedback connections) which is equivalent to a Bayesian network combining bottom-up likelihoods with top-down priors to compute the maximum *a posteriori* (MAP) estimate. Here, the priors are embedded in the weight

structure of the network. In simulations, we include an additional scalar $k_{td}$ that sets the relative weighting of bottom-up versus top-down signals.

$$\dot{\mathbf{x}} = A(\mathbf{u} - A^T\mathbf{x}) - k_{td}(\mathbf{x} - B^Ty) - \mathbf{x}/\tau \tag{9}$$

## Error signals computed in the feedback model

In *Equation (9)*, inference can be thought of as proceeding through integration of inputs on the dendrites of neuron population *x*. In this scenario, all computations are implicit in dendritic integration. Alternatively, the computations in *Equation (9)* can be done in two steps where, in the first step, reconstruction errors are computed (i.e. $e_0 = u\text{-}A^Tx$, $e_1 = x\text{-}B^Ty$) and explicitly represented in a separate error coding population. These error signals can then be integrated by their downstream target population to generate the requisite update to the state signal of neuron population *x*.

$$\dot{\mathbf{x}} = A\mathbf{e}_0 - k_{td}\mathbf{e}_1 - \mathbf{x}/\tau, \quad \dot{\mathbf{y}} = B\mathbf{e}_1 - y/\tau \tag{10}$$

An advantage of this strategy is that the a state unit now directly receives errors as inputs, and those inputs allow implementation of an efficient Hebbian rule for learning weight matrices (*Rao and Ballard, 1999*) – the gradient rule for learning is simply a product of the state activation and the input error activation (weight updates obtained by differentiating *Equation (6)* with respect to weight matrices *A* and *B*: $\Delta A = x \bullet e_0^T$, $\Delta A^T = e_0 \bullet x^T$, $\Delta B = y \bullet e_1^T$, and $\Delta B^T = e_1 \bullet y$). Thus, the reconstruction errors serve as computational intermediates for both the gradients of online inference mediated by dendritic integration (dynamics in state space, *Equation (10)*) and gradients for offline learning mediated by Hebbian plasticity (dynamics in weight space).

In order for the reconstruction errors at each layer to be scaled appropriately in the feedback model, we invoke an additional downstream variable *z* to predict activity at the top stage such that, instead of $e_2 = y$ which scales as a state variable, we have $e_2 = y\text{-}C^Tz$ (*Figure 5A*). This overall model reflects a state and error coding model as opposed to a state only model.

## Feedback (three-stage)

For the simulations in *Figure 5*, three-stage versions of the above equations were used. These deeper networks were also wider such that they began with four input units (*u*) instead of only two inputs in the two-stage models. These inputs converged through successive processing stages (*w,x, y*) to one unit at the top node (*z*) (*Figure 5*).

## Feedback (nonlinear reconstruction)

To test the generality of our findings beyond a linear reconstruction cost, we simulated feedback-based models which optimized different candidate cost functions proposed for the ventral stream (*Figure 6*). In nonlinear hierarchical inference, reconstruction is performed using a monotonic nonlinearity with a threshold (*th*) and bias (*bi*):

$$c = \frac{1}{2}(\mathbf{u} - f(A^T\mathbf{x}))^2 + \frac{1}{2}(\mathbf{u} - f(B^Ty))^2, where\, f(x) = \tanh(x - th) + bi \tag{11}$$

$$\dot{\mathbf{x}} = A(\mathbf{u} - f(A^T\mathbf{x}))(1 - \tanh(A^T\mathbf{x} - th)^2) - k_{td}(\mathbf{x} - f(B^Ty)) - \mathbf{x}/\tau$$
$$\dot{\mathbf{y}} = B(\mathbf{x} - f(B^T\mathbf{y}))(1 - \tanh(B^T\mathbf{y} - th)^2) - \mathbf{y}/\tau \tag{12}$$

## Feedback (linear construction)

Instead of a reconstruction cost where responses match the input (i.e. generative model) as in unsupervised learning, we additionally simulated the states and errors in a feedback network minimizing a linear construction cost where the network is producing responses to match a given output (i.e. discriminative model) similar to supervised learning:

$$C = \frac{1}{2}(A\mathbf{u} - \mathbf{x})^2 + \frac{1}{2}(B\mathbf{x} - \mathbf{y})^2 \tag{13}$$

$$\dot{\mathbf{x}} = (A\mathbf{u} - \mathbf{x}) - k_{td}B^T(B\mathbf{x} - \mathbf{y}) - x/\tau, \quad \dot{\mathbf{y}} = (B\mathbf{x} - \mathbf{y}) - \mathbf{y}/\tau \tag{14}$$

## Model simulation

To simulate the dynamical systems in *Equations (2)-(14)*, a step input *u* was applied. This input was smoothed using a Gaussian kernel to approximate the lowpass nature of signal propagation in the series of processing stages from the retina to pIT:

$$\mathbf{u}(t) = \frac{1}{\sqrt{2\pi\sigma^2}} e^{\frac{-(t-t_0)^2}{2\sigma^2}} * \mathbf{h}(t) \Rightarrow \dot{\mathbf{u}} = \frac{1}{\sqrt{2\pi\sigma^2}} e^{\frac{-(t-t_0)^2}{2\sigma^2}} \cdot \mathbf{h} \tag{15}$$

where the elements of *h* are scaled Heaviside step functions. The input is thus a sigmoidal ramp whose latency to half height is set by $t_0$ and rise time is set by σ. For simulation of two-stage models, there were ten basic parameters: latency of the input $t_0$, standard deviation of the Gaussian ramp σ, system time constant τ, input connection strength *a*, feedforward connection strength *b*, the four input values across two stimulus conditions (i.e. $h_{11}$, $h_{12}$, $h_{21}$, $h_{22}$), and a factor *sc* for scaling the final output to the neural activity. In the deeper three-stage network, there were a total of fifteen parameters which included an additional feedforward connection strength *c* and additional input values since the three-stage model had four inputs instead of two. The lateral inhibition model class required one additional parameter $k_l$ as did the normalization model family $k_s$, and for feedback model simulations, there was an additional feedback weight $k_{td}$ to scale the relative contribution of the top-down errors in driving online inference. For the error coding variants of the feedback model, gain parameters *c* (two-stage) and *d* (three-stage) were included to scale the overall magnitude of the top level reconstruction error.

## Model parameter fits to neural data

In fitting the models to the observed neural dynamics, we mapped the summed activity in the hidden stage (*w*) to population averaged activity in pIT, and we mapped the summed activity in the output stage (*y*) to population averaged signals measured in aIT. To simulate error coding, we mapped the reconstruction errors $e_1 = w\text{-}B^T x$ and $e_3 = y\text{-}C^T z$ to activity in pIT and aIT, respectively. We applied a squaring nonlinearity to the model outputs as an approximation to rectification since recorded extracellular firing rates are non-negative (and linear rectification is not continuously differentiable). Analytically solving this system of dynamical *Equations (2)-(15)* for a step input is precluded because of the higher order interaction terms (the roots of the determinant and hence the eigenvalues/eigenvectors of a 3 × 3 or larger matrix are not analytically determined, except for the purely feedforward model which only has first-order interactions), and in the case of the normalization models, there is an additional nonlinear dependence on the shunt term. Thus, we relied on computational methods (constrained nonlinear optimization) to fit the parameters of the dynamical systems to the neural data with a quadratic (sum of squares) loss function.

Parameter values were fit in a two-step procedure. In the first step, we fit only the difference in response between image classes (differential mode which is the selectivity profile over time, see *Figure 5A*, right data panel), and in the second step, we refined fits to capture an equally weighted average of the differential mode and the common mode (the common mode is the average across images of the response time course of visual drive). This two-step procedure was used to ensure that each model had the best chance of fitting the dynamics of selectivity (differential mode) as these selectivity profiles were the main phenomena of interest but were smaller in size (20% of response) compared to overall visual drive. In each step, fits were done using a large-scale algorithm (interior-point) to optimize coarsely, and the resulting solution was used as the initial condition for a medium-scale algorithm (sequential quadratic programming) for additional refinement. The lower and upper parameter bounds tested were: $t_0$=[50 70], σ=[0.5 25], τ =[0.5 1000], $k_l,k_s,k_{td}$=[0 1], *a,b,c, d*=[0 2], *h*=[0 20], *sc*=[0 100], *th*=[−20 20], and *bi*=[−1 1] which proved to be adequately liberal as parameter values converged to values that did not generally approach these boundaries. To avoid local minima, the algorithm was initialized to a number of randomly selected points (n = 50), and after fitting the differential mode, we took the top fits (n = 25) for each model class and used these as initializations in subsequent steps. The single best fitting instance of each model class is shown in the main figures.

## Data and code availability

Source data including all image stimuli and neural data are available online in accompanying files. Complete model code is also available in accompanying files online. All data analysis and computational modeling were done using custom scripts written in MATLAB.

## Statistics

Error bars represent standard errors of the mean obtained by bootstrap resampling (n = 1000). All statistical comparisons including those of means or correlation values were obtained by bootstrap resampling (n = 1000) producing p-values at a resolution of 0.001 so that the lowest p-value that can be reported is p = 0.000 given the resolution of this statistical analysis. All statistical tests were two-sided unless otherwise specified. Spearman's rank correlation coefficient was used.

## Acknowledgements

We thank J Deutsch, K Schmidt, and P Aparicio for help with MRI and animal care and B Andken and C Stawarz for help with experiment software.

## Additional information

### Funding

| Funder | Grant reference number | Author |
| --- | --- | --- |
| National Institutes of Health | R01-EY014970 | James J DiCarlo |
| National Institutes of Health | K99-EY022671 | Elias B Issa |
| National Institutes of Health | F32-EY019609 | Elias B Issa |
| National Institutes of Health | F32-EY022845 | Charles F Cadieu |
| Office of Naval Research | MURI-114407 | James J DiCarlo |
| MIT McGovern Institute for Brain Research | | James J DiCarlo |

The funders had no role in study design, data collection and interpretation, or the decision to submit the work for publication.

### Author contributions

Elias B Issa, Conceptualization, Formal analysis, Funding acquisition, Investigation, Visualization, Methodology, Writing—original draft, Writing—review and editing; Charles F Cadieu, Conceptualization, Formal analysis, Methodology, Writing—original draft; James J DiCarlo, Conceptualization, Resources, Supervision, Funding acquisition, Writing—original draft, Project administration, Writing—review and editing

### Author ORCIDs

Elias B Issa (iD) http://orcid.org/0000-0002-5387-7207

### Ethics

Animal experimentation: All procedures were performed in compliance with National Institutes of Health guidelines and the standards of the MIT Committee on Animal Care (IACUC protocol #0111-003-14) and the American Physiological Society.

### Decision letter and Author response

Decision letter https://doi.org/10.7554/eLife.42870.011
Author response https://doi.org/10.7554/eLife.42870.012

# Additional files

## Supplementary files

• Source data 1. Source neural data for figures 1-5 and model code for figures 5-6.
DOI: https://doi.org/10.7554/eLife.42870.008
• Transparent reporting form
DOI: https://doi.org/10.7554/eLife.42870.009

## Data availability

All data generated or analyzed during this study are included in the supporting files for the manuscript. Source data files have been provided for Figures 1-5, and code for computational models in Figures 5 & 6 is provided.

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
