## [Decision Letter]

[Editors’ note: a previous version of this study was rejected after peer review, but the authors submitted for reconsideration. The first decision letter after peer review is shown below.]

Thank you for submitting your work entitled "Neural dynamics at successive stages of the ventral visual stream are consistent with hierarchical error signals" for consideration by eLife. Your article has been reviewed by three peer reviewers, and the evaluation has been overseen by a Reviewing Editor and a Senior Editor.

Our decision has been reached after consultation between the reviewers. Based on these discussions and the individual reviews below, we regret to inform you that your submission has been rejected for publication in eLife at this time. However, we would be willing to consider a resubmission that addresses the suggestions and concerns outlined below.

Reviewers felt that the results have strong potential importance for understanding the role of feedback in ventral pathway, on a par with Schwiedrzik and Freiwald, 2017. Schwiedrzik and Freiwald show prediction-based modulation based on extensive training with arbitrary pairings of two successive faces. Your results show prediction-based modulation of a very different kind, based on mismatch between the driving stimulus (for this area, a contralateral eye) and surrounding stimulus patches that don't fit the expectation for a normal face. This is prediction based on spatial context rather than temporal training, and in that sense a more natural example that seems more relevant to normal ventral pathway processing and the idea that feedforward/feedback interactions progressively refine the estimate of what is present in the world. Both papers are important for being initial demonstrations that feedback in the ventral pathway may modulate signals based on high-level predictions, a longstanding theory.

The strongest concern was about the interpretation of the results as a switch in coding value from face to anti-face, because the neurons are still responding, at only a slightly different level, to an image with their primary driving stimulus, a contralateral eye, which can drive responses by itself. A small change in relative responses to very similar stimuli does not mean that coding polarity has suddenly flipped. Predictive coding theory does not involve neurons changing the meaning of their state value signals. The distinction made here between "faces" and "anti-faces" is an arbritrary boundary in a small set of stimuli all containing at least a contra-lateral eye, the overall shape of a face, and face-like texture. There is no demonstration that these neurons begin responding to actual non-face stimuli, i.e. other objects with completely different shapes and appearances (which were part of the stimulus set and could be compared). The example stimuli do not even exhibit the switch since they start out on the non-face side of the boundary. The d' between "faces" and "anti-faces" is only around 0.1 or 0.2.

Instead, the results have a simple interpretation based on the Issa and Dicarlo 2012 study of face responses in pIT. That study demonstrated that early, feedforward pIT face responses are driven by images of an eye, positioned near the contralateral top of a rounded outline. The relatively stronger responses to the "non-face" stimuli, which contain the contralateral eye but with missing or misplaced other face parts, make sense as a positive modulation of a state signal for the contralateral eye when the eye is unpredictable/surprising based on the spatial surround being inconsistent (jumbled or absent face features). This is analogous to the increase in responses to unexpected face identities/orientations observed by Schwiedrzik and Freiwald.

We suggest a resubmission based on this simpler interpretation. In addition, as elaborated in the separate reviews, a resubmission should address strong concerns about conclusions from the modeling analyses, especially the claim to support theories involving pure error signals, and claims to explain many other general ventral pathway phenomena in terms of this one observation.

*Reviewer #1:*

Issa et al. present an analysis of response dynamics at three stages in the monkey face patch system. The stimuli are selected from the face part rearrangements used in the Issa and Dicarlo 2012 study of face responses in pIT. That study demonstrated that early, feedforward pIT face responses are driven primarily by images of an eye, positioned near the contralateral top of a rounded outline. Here, they compare responses in early (60-90 ms after onset) and late (100-130 ms after onset) response phases, for two types of stimuli (all of which contained at least one eye, at or near the optimum position for driving pIT responses), "faces" (which here means eight stimuli in which multiple other face parts are in their correct locations) and "non-faces" (which here means stimuli in which most other face parts besides the contralateral eye are missing or misplaced). All of these stimuli drove strong initial responses from pIT neurons, presumably because they all contained an eye at or near the critical location, consistent with Issa and Dicarlo 2012. Responses to the "face" stimuli declined by 18% in the 100-130 ms period relative to the 60-90 ms period. In contrast, responses to the "non-face" stimuli did not decline.

This result has a straightforward interpretation in terms of predictive coding theory, in which neural signals reflect, at least in part, deviations of feedforward stimulus signals from predictions based on temporal history, spatial surround information, or other factors. Retinal ganglion cells manifest both, responding either to increased or to decreased luminance at their receptive field center, relative to the previous luminance value at the receptive field center and relative to the luminance value in the spatial surround. Schwiedrzik and Freiwald, 2017, recently demonstrated predictive coding based on temporal history in face patch ML (probably corresponding to the pIT neurons in this new manuscript). Monkeys were passively exposed to many repeats of 9 specific pairings of face images, in which both identity and head orientation varied. After training, ML neurons were tested with both the trained pairings and with novel pairings of the same initial and successor stimuli. As expected, neurons exhibited different preferences for stimuli. Responses to preferred successor stimuli were about 17% higher when the preceding stimulus was switched and therefore predicted a different head orientation, face identity, or both. These three conditions evoked approximately equal deviations in the 120-210 ms period after onset, but at 300-440 ms differences in identity caused the largest deviations (around 20%) and differences in head orientation had no effect. This presumably reflects the more gradual evolution of identity information, which is represented in the most anterior face patch AM. The sensitivity of ML to identity errors, even when head orientation was identical, reflects the top-down origin of prediction signals.

The main result in this new Issa et al. manuscript seems to be a similar-sized prediction error based on surround information rather than temporal history. Later responses in pIT are about 18% lower when the information surrounding the eye stimulus is consistent with a face (thus no prediction error) than when the surrounding information is inconsistent with a face, so that the presence of the eye by itself becomes a more unexpected, unpredictable stimulus element. The appearance of this difference only after the initial pIT response (60-90 ms) is evidence that higher-level processing of the other, surrounding face elements was required to produce the prediction error difference. A similar pattern is observed for some cIT neurons, while aIT neurons maintain their selectivity for the face stimuli, providing the likely top-down source for prediction signals.

However, the authors interpret their phenomenon not in terms of prediction modulation of the feedforward signals for a contralateral eye stimulus from pIT but in terms of selectivity for faces in general, claiming that pIT neurons "decreased their preference for faces, becoming anti-face preferring on average", which is diagrammed in Fig. 1B. This is confusing for a number of reasons. First, theories of prediction error coding do not involve neurons suddenly changing the basic meaning of their signals. Second, the "anti-face" preference is really a slightly differential response to the "face" stimuli vs. the "non-face" partially scrambled stimuli used here, all of which have the contralateral eye, which these authors have previously shown is the primary feedforward signal from pIT (Issa and Dicarlo, 2012). The main result is more naturally interpreted as a continuing response to the contralateral eye, modulated by the surrounding face-part information, in a way consistent with predictive coding. Slightly differential responses to the two groups of stimuli here (a d' of around 0.1 or 0.2) do not signify a reversal of preference for faces. There is no demonstration that pIT neurons suddenly respond more strongly to toasters and hammers than to faces. Third, there is no reason to think that the most informative tuning dimension for these neurons is a continuum between faces and other objects. Throughout the face patch system, neurons are tuned for the details of facial structure, and, consistent with this, Schwiedrzik and Freiwald report that error coding in ML/pIT occurs along dimensions like head orientation and identity, as one would expect. Thus the Fig. 1B diagram, and the logic throughout the text and the figures, which explain prediction errors in terms of reversing face selectivity, is confusing (i) because the face patch system does not simply discriminate faces from other objects, (ii) because wholesale tuning reversal is not expected based on prediction coding theory, and (iii) a small change in relative responses to very similar stimuli does not mean that coding polarity in general has suddenly flipped.

The second part of the paper is an extended analysis of predictive coding models. This part of the paper also suffers from the confusing interpretation that pIT neurons reverse their tuning polarity. The authors show that only models capable of computing errors between layers can fit their results, which makes sense. They claim to explain a number of previously observed phenomena with their model, including sublinear part integration, faster attenuation of signals for familiar stimuli, and ramp coding of face structure. These extrapolations to phenomena well beyond the scope of this paper seem qualitative, tacked on, and unconvincing. The authors also make a comparison of dynamics in superficial and deep neurons to network dynamics of "state" and "error" neurons, and claim that superficial, forward projecting neurons have longer dynamics, like state neurons, and thus state signals, not error signals, are propagated forward, consistent with error back propagation and not predictive coding. This also seems tenuous, speculative, and beyond the scope of this paper. If the authors wanted to explore this idea, they would need among other things to distinguish feedforward input layer 4 from deep and superficial layers and characterize differences in stimulus coding, error coding, and dynamics in all layers.

In summary, the basic error-related phenomenon makes sense in terms of the contralateral eye coding role for pIT neurons previously established by these authors, and this constitutes an important new finding that extends understanding of top-down feedback effects and their possible relation to error coding in ventral pathway vision. The phenomenon does not make very clear sense in terms of face/non-face coding. The manuscript would benefit from clarifying this interpretation, reducing the attempts to extrapolate the results so far through modeling, and doing more to situate the results with respect to the existing literature, especially the extremely relevant paper by Schwiedrzik and Freiwald, which receives only a glancing reference here.

*Reviewer #2:*

This paper examines the firing rate dynamics of neurons in three hierarchically arranged face sensitive areas of the ventral macaque monkey cortex, areas and neurons proposed to be responsible for face vs. non-face detection behavior. The key empirical finding is that despite early selectivity for faces in neurons in lower areas, preference for faces decreases within several 10s of milliseconds, while at the same time face preference in the top area remains high. The authors interpret their finding as consistent with neural circuitry that computes an error signal measuring the difference between the initial input activity and subsequent feedback signal from the higher area(s). This interpretation is consistent with previous theories of predictive coding, in contrast to other theories that predict no decrease in the selectivity of earlier neurons. About half of the paper is then devoted to showing results from several classes of simplified dynamical models, and to comparing the pattern of model results with the empirical observations. The paper addresses a significant and deep question as to the computational functions of inter-area visual cortical circuitry. The empirical results are new, and the modeling has a number of novel features that should be of broad interest, such as the quantitative analysis of dynamic feedback networks that include both state and error units.

The scope of the paper, empirical and theoretical, results in a fairly long read. Some of length could benefit from reducing redundant descriptions of the classes of previous models within the paper. But the paper is also missing details in a few parts. The comments below primarily have to do with the need to fill in missing explanations in the data analysis, to better organize model descriptions in the early parts of the paper, and improve the organization.

- The Materials and methods goes into detail regarding the various stimulus variants. But other than the numbers of presentations and their durations, the details of the temporal presentation of stimulus types was not specified. The manuscript should better describe how the neural spiking data was analyzed with respect to the stimulus sequences, e.g. how the data in Figure 2 was determined. The lack of detail makes it difficult to evaluate possible interactions between stimulus type and presentation timing. It would also help to better describe how the relative timing in responses between the areas was determined, and its variability over sites.

- It seems odd that the laminar-related hypotheses first appear at the end of the computational section. It would be more consistent with the organization of the rest of the paper if the hypotheses were at the beginning (in the context of predictive coding, etc.). This section also seems isolated in that the reader is left hanging with the evidence that superficial units behave like state rather than error units. It would help to have some discussion of other literature, such as laminar analysis of fMRI data in human V1 that has been interpreted in terms of predictive coding.

*Reviewer #3:*

The paper presents neurophysiological data from macaque suggesting a set of PIT and CIT neurons might be explicitly coding prediction error signals specifically to face configuration. The paper also investigated a set of models and argued that the PIT and CIT error responses can only be explained by a recurrent feedback model implementing predictive coding. Overall, the paper is reasonably well-written with interesting data and provocative claims but I have three concerns as to whether the claims are fully supported by the data. First, while the effect observed in PIT might indeed be mediated by feedback, horizontal (intra-areal interactIon) mechanisms cannot be ruled out empirically. This is because the error-computing "model" can be implemented within each cortical area as well. Some additional analysis of the data based on timing of the different effects in PIT and AIT might help argue for feedback over horizontal interaction. Second, the empirical observations seem to be also consistent with the simpler notion that PIT signals are coding face parts, and that these face part state variables were then amplified by the attention due to part-whole incongruence. Thus, the signals might not be pure error signals and the data might not be sufficient for arguing for the predictive coding model, which projects only error signals in feedforward connections, over other theories, which project attentional modulated state variables. Third, none of the PIT examples in Figure 2 showed preferred face configuration over non-face configurations in the early response window as Figure 3's graphs indicated and the text suggested. The reversal of neuronal preference from face in early responses to non-face in late responses was supposed to be key evidence in support of the idea of the error signals, and should be demonstrated in Figure 2. These concerns should be addressable by revising the claims appropriately, adding cautionary notes/caveats in the interpretation of the data, or maybe providing some additional analysis, graphs and clarification.

Elaboration:

The main claim was that many "face-selective" neurons in PIT (less so in CIT, and not in AIT) responded more to face images in normal configuration than to images of face components in unusual configuration (non-face stimuli) in the 60-100 ms window post-stimulus onset and then changed their preference to non-face in the 100-130 ms window. The evidence presented in the d' graphs in Figure 3 was rather compelling, and so were the simulation results of the models (Fig. 6 and 7). However, none of the PIT example cells or sites as shown in Figure 2 actually exhibited a preference for the normal face configurations in the early response. The examples showed that the neurons tended to prefer non-face images even in the early response window such preference might become more accentuated in the later part of the responses, which was different from the simulation results. It would be more compelling to provide some examples in Figure 2 that are consistent with the key claim.

Given the PIT neurons responded more to the non-faces from the beginning in Figure 2, it is important to confirm that these neurons actually were face or face part preferring or selective, by showing their responses to house, body parts and places as well, which were also tested. Assuming that these neurons indeed are face part sensitive, the data did suggest that the responses to face parts were stronger when they were part of an incoherent face configuration, rather than a part of a coherent face configuration. The stronger responses to the non-face images could be error signals between the expectation of the global configuration and local patterns but they might not be pure error signals, but attentional signals drawn by the errors to enhance the saliency of feature responses. It is not clear whether the neural responses could be considered as "error signals", or simply the state variables modulated by the error signals.

The models do not seem to provide many new insights or predictions. By design, an error computing architecture will show error signals. It is obvious that simple lateral inhibition or normalization could not produce the effects, but I am not convinced that the observed effects cannot be explained by horizontal interaction within each area. The paper did not present experimental evidence establishing that the prediction error signals were mediated by feedback from AIT, rather than by horizontal connections within PIT. Short of reversible deactivating IT or computing conditional Granger causality using simultaneously recordings of AIT and PIT sites, any conclusion on feedback however can only be tentative based on current evidence on single unit recording. The authors might be able to provide some evidence based on relative timing of the responses in AIT and PIT to argue for this point. Figure 2's examples did suggest that AIT neurons' face preference might precede PIT's non-face preference, which might reflect a potential causal relationship. It would be prudent to point out that although simple lateral inhibition and normalization, as explored by the authors, might not explain the error signals, the prediction errors can easily be calculated within PIT with horizontal connections. After all, a significant number of the PIT neurons also preferred face over non-faces as in AIT. Besides, all the contextual modulation effects (end-stopping, surround suppression) that were cited in Rao and Ballard's paper could also be implemented with horizontal connections in V1. Thus, the modeling effort could only be used to argue that there were the error signals between global versus local representations were computed in PIT, it could not be used as a "proof" for the involvement of a feedback mechanism. So it might be prudent to be more cautious about the conclusions that can be drawn from the modeling effort.

Showing error signals (between face parts and global face configuration) in PIT but not AIT is interesting. This is consistent with earlier observation, for example Issa's earlier study, that PIT neurons are encoding face parts and AIT neurons are encoding the whole faces. If PIT neurons are encoding face parts, reflexive attention and vigilance will naturally be devoted to the neurons coding the parts due to the inconsistency between local parts and global configuration. Thus, it could be an error-induced attentional effect rather than the error signal itself.

The idea of predictive coding or residue coding was first proposed by Mumford (Biological Cybernetics 1993). In its purest form, Mumford's idea is a generalization of Barlow's sparse coding idea to the entire visual hierarchy, and is similar conceptually to Burt and Adelson's Laplacian pyramid, with different areas coding only the "residue signals". Rao and Ballard proposed a Kalman filter implementation of the idea but with a significant modification (and retreat) - reintroducing the state variables into each area in addition to the residue signals. Thus, experimentally, it is difficult to distinguish the classical idea of interactive activation (McClelland and Rumelhart) or adaptive resonance (Grossberg) or hierarchical Bayes (Lee and Mumford) from Rao and Ballard's Kalman filter model because all these models require both state and error representations in each layer. Experimentally the only measurable difference to distinguish the "predictive coding" model from the other models is that its feedforward signals from one visual area to another area contain *only* the error signals in the predictive coding model, while the other models will project the "state variables" or beliefs, possibly modulated by attention. Thus, the data presented in this paper could not be used to distinguish whether the neural signals were the error (or residue signals or Bayesian surprises), or state variables or beliefs enhanced by attention. The data only showed that the incongruence between the local and the global representations lead to enhancement of the face part responses of the PIT neurons. The distinction is subtle but important. The existence of error related signals was well-known through the cortex, but they cannot be considered as conclusive evidence for supporting Mumford or the Rao-Ballard's hierarchical predictive coding theory.

---

## [Author Response]

[Editors’ note: the author responses to the first round of peer review follow.]

Thank you for submitting your work entitled "Neural dynamics at successive stages of the ventral visual stream are consistent with hierarchical error signals" for consideration by eLife. Your article has been reviewed by three peer reviewers, and the evaluation has been overseen by a Reviewing Editor and a Senior Editor.Our decision has been reached after consultation between the reviewers. Based on these discussions and the individual reviews below, we regret to inform you that your submission has been rejected for publication in eLife at this time. However, we would be willing to consider a resubmission that addresses the suggestions and concerns outlined below.Reviewers felt that the results have strong potential importance for understanding the role of feedback in ventral pathway, on a par with Schwiedrzik and Freiwald, 2017. Schwiedrzik and Freiwald show prediction-based modulation based on extensive training with arbitrary pairings of two successive faces. Your results show prediction-based modulation of a very different kind, based on mismatch between the driving stimulus (for this area, a contralateral eye) and surrounding stimulus patches that don't fit the expectation for a normal face. This is prediction based on spatial context rather than temporal training, and in that sense a more natural example that seems more relevant to normal ventral pathway processing and the idea that feedforward/feedback interactions progressively refine the estimate of what is present in the world. Both papers are important for being initial demonstrations that feedback in the ventral pathway may modulate signals based on high-level predictions, a longstanding theory.

We thank the editors and reviewers for this positive comparison of our work to recently published work. In the Discussion, we have included a more direct comparison to the Neuron 2017 study highlighting points which make our study novel relative to this recent work.

The strongest concern was about the interpretation of the results as a switch in coding value from face to anti-face, because the neurons are still responding, at only a slightly different level, to an image with their primary driving stimulus, a contralateral eye, which can drive responses by itself. A small change in relative responses to very similar stimuli does not mean that coding polarity has suddenly flipped.

We have revised our phrasing to avoid implying that neurons prefer non-faces. Rather, we state throughout the text that pIT neuron responses are modulated by atypical face-part configurations. In other words, they are tuned to the local features (e.g. eye features), but signal an error in the configurational context with respect to that local feature preference. We also clarify that the observed dynamics reflect a relatively small change modulating the overall strong response for face features.

Predictive coding theory does not involve neurons changing the meaning of their state value signals. The distinction made here between "faces" and "anti-faces" is an arbritrary boundary in a small set of stimuli all containing at least a contra-lateral eye, the overall shape of a face, and face-like texture. There is no demonstration that these neurons begin responding to actual non-face stimuli, i.e. other objects with completely different shapes and appearances (which were part of the stimulus set and could be compared). The example stimuli do not even exhibit the switch since they start out on the non-face side of the boundary. The d' between "faces" and "anti-faces" is only around 0.1 or 0.2.Instead, the results have a simple interpretation based on the Issa and Dicarlo 2012 study of face responses in pIT. That study demonstrated that early, feedforward pIT face responses are driven by images of an eye, positioned near the contralateral top of a rounded outline. The relatively stronger responses to the "non-face" stimuli, which contain the contralateral eye but with missing or misplaced other face parts, make sense as a positive modulation of a state signal for the contralateral eye when the eye is unpredictable/surprising based on the spatial surround being inconsistent (jumbled or absent face features). This is analogous to the increase in responses to unexpected face identities/orientations observed by Schwiedrzik and Freiwald.We suggest a resubmission based on this simpler interpretation. In addition, as elaborated in the separate reviews, a resubmission should address strong concerns about conclusions from the modeling analyses, especially the claim to support theories involving pure error signals, and claims to explain many other general ventral pathway phenomena in terms of this one observation.

We appreciate the reviewer’s concerns that the modeling section extrapolates far from the data. We have significantly curtailed the modeling and removed the claims regarding phenomena outside the scope of the main dataset. As a result, the manuscript is significantly shorter, by ~25%, in text and figures. We hope this more concise manuscript reads more easily and now makes only the fully supported points.

Reviewer #1:[…] However, the authors interpret their phenomenon not in terms of prediction modulation of the feedforward signals for a contralateral eye stimulus from pIT but in terms of selectivity for faces in general, claiming that pIT neurons "decreased their preference for faces, becoming anti-face preferring on average", which is diagrammed in Fig. 1B. This is confusing for a number of reasons. First, theories of prediction error coding do not involve neurons suddenly changing the basic meaning of their signals. Second, the "anti-face" preference is really a slightly differential response to the "face" stimuli vs. the "non-face" partially scrambled stimuli used here, all of which have the contralateral eye, which these authors have previously shown is the primary feedforward signal from pIT (Issa and Dicarlo, 2012). The main result is more naturally interpreted as a continuing response to the contralateral eye, modulated by the surrounding face-part information, in a way consistent with predictive coding. Slightly differential responses to the two groups of stimuli here (a d' of around 0.1 or 0.2) do not signify a reversal of preference for faces. There is no demonstration that pIT neurons suddenly respond more strongly to toasters and hammers than to faces. Third, there is no reason to think that the most informative tuning dimension for these neurons is a continuum between faces and other objects. Throughout the face patch system, neurons are tuned for the details of facial structure, and, consistent with this, Schwiedrzik and Freiwald report that error coding in ML/pIT occurs along dimensions like head orientation and identity, as one would expect. Thus the Fig. 1B diagram, and the logic throughout the text and the figures, which explain prediction errors in terms of reversing face selectivity, is confusing (i) because the face patch system does not simply discriminate faces from other objects, (ii) because wholesale tuning reversal is not expected based on prediction coding theory, and (iii) a small change in relative responses to very similar stimuli does not mean that coding polarity in general has suddenly flipped.

We have changed our phrasing throughout the text to indicate that pIT neurons are tuned to face parts but express a modulatory signal for a mismatched face configuration. We also remove any reference implying reversing selectivity for part configuration. Rather, we are simply claiming that there is a higher response for atypical configurations in the late response phase. We have updated the predictions Figure 1B accordingly.

The second part of the paper is an extended analysis of predictive coding models. This part of the paper also suffers from the confusing interpretation that pIT neurons reverse their tuning polarity. The authors show that only models capable of computing errors between layers can fit their results, which makes sense. They claim to explain a number of previously observed phenomena with their model, including sublinear part integration, faster attenuation of signals for familiar stimuli, and ramp coding of face structure. These extrapolations to phenomena well beyond the scope of this paper seem qualitative, tacked on, and unconvincing.

We agree with the reviewer and have removed these modeling analyses which greatly simplify the manuscript without losing the main point of the work.

The authors also make a comparison of dynamics in superficial and deep neurons to network dynamics of "state" and "error" neurons, and claim that superficial, forward projecting neurons have longer dynamics, like state neurons, and thus state signals, not error signals, are propagated forward, consistent with error back propagation and not predictive coding. This also seems tenuous, speculative, and beyond the scope of this paper. If the authors wanted to explore this idea, they would need among other things to distinguish feedforward input layer 4 from deep and superficial layers and characterize differences in stimulus coding, error coding, and dynamics in all layers.In summary, the basic error-related phenomenon makes sense in terms of the contralateral eye coding role for pIT neurons previously established by these authors, and this constitutes an important new finding that extends understanding of top-down feedback effects and their possible relation to error coding in ventral pathway vision. The phenomenon does not make very clear sense in terms of face/non-face coding. The manuscript would benefit from clarifying this interpretation, reducing the attempts to extrapolate the results so far through modeling, and doing more to situate the results with respect to the existing literature, especially the extremely relevant paper by Schwiedrzik and Freiwald, which receives only a glancing reference here.

In addition to clarifying the nature of the neural phenomenon and shortening the modeling section, we relate our work to Schwiedrzik and Freiwald, 2017, highlighting how our study extends that work by recording from the putative areas providing prediction signals while also exposing a spatial prediction rather than a temporal prediction requiring behavioral training.

Reviewer #2:[…] The paper addresses a significant and deep question as to the computational functions of inter-area visual cortical circuitry. The empirical results are new, and the modeling has a number of novel features that should be of broad interest, such as the quantitative analysis of dynamic feedback networks that include both state and error units.The scope of the paper, empirical and theoretical, results in a fairly long read. Some of length could benefit from reducing redundant descriptions of the classes of previous models within the paper.

We have simplified our explanation of the modeling section presenting a feedforward model as well as the predictive coding model only and removing the model prediction section. The paper is much shorter by ~25% in text and figure material.

But the paper is also missing details in a few parts. The comments below primarily have to do with the need to fill in missing explanations in the data analysis, to better organize model descriptions in the early parts of the paper, and improve the organization.- The Materials and methods goes into detail regarding the various stimulus variants. But other than the numbers of presentations and their durations, the details of the temporal presentation of stimulus types was not specified. The manuscript should better describe how the neural spiking data was analyzed with respect to the stimulus sequences, e.g. how the data in Figure 2 was determined. The lack of detail makes it difficult to evaluate possible interactions between stimulus type and presentation timing. It would also help to better describe how the relative timing in responses between the areas was determined, and its variability over sites.

We thank the reviewer for this suggestion as it is important to discuss the order and timing of presentation in our stimuli for interpreting whether the results could be the result of attention to novel faces. We now discuss this in the Results and Materials and methods sections to clarify that stimuli (normal vs. novel faces) were randomly interleaved, minimizing any possibility of attention or priming signals to the novel facelike images.

- It seems odd that the laminar-related hypotheses first appear at the end of the computational section. It would be more consistent with the organization of the rest of the paper if the hypotheses were at the beginning (in the context of predictive coding, etc.). This section also seems isolated in that the reader is left hanging with the evidence that superficial units behave like state rather than error units. It would help to have some discussion of other literature, such as laminar analysis of fMRI data in human V1 that has been interpreted in terms of predictive coding.Reviewer #3:The paper presents neurophysiological data from macaque suggesting a set of PIT and CIT neurons might be explicitly coding prediction error signals specifically to face configuration. The paper also investigated a set of models and argued that the PIT and CIT error responses can only be explained by a recurrent feedback model implementing predictive coding. Overall, the paper is reasonably well-written with interesting data and provocative claims but I have three concerns as to whether the claims are fully supported by the data. First, while the effect observed in PIT might indeed be mediated by feedback, horizontal (intra-areal interactIon) mechanisms cannot be ruled out empirically. This is because the error-computing "model" can be implemented within each cortical area as well. Some additional analysis of the data based on timing of the different effects in PIT and AIT might help argue for feedback over horizontal interaction.

We have expanded on how the observed phenomenon may or may not constrain underlying mechanisms. First, we did provide analyses of how the timing of information in cIT and aIT relates to the late signals in pIT, and this analysis was consistent with a potential feedback interaction (early cIT/aIT response predicts late pIT response pattern). This was in the original manuscript but is highlighted more strongly since it is important for supporting a potential feedback mechanism. Second, in the Results and Discussion modeling sections, we make sure to state the limits of our observations for determining whether within area (lateral) recurrence or top-down (feedback) recurrence are the actual underlying mechanism and propose future causal studies to test these putative mechanisms for generating the observed neural phenomena.

Second, the empirical observations seem to be also consistent with the simpler notion that PIT signals are coding face parts, and that these face part state variables were then amplified by the attention due to part-whole incongruence. Thus, the signals might not be pure error signals and the data might not be sufficient for arguing for the predictive coding model, which projects only error signals in feedforward connections, over other theories, which project attentional modulated state variables.

The observed responses were unlikely to result from attention signals. The stimuli were randomly interleaved so that the subject had no prior expectations of the stimulus class (normal versus altered face-part configuration). Furthermore, the IT neural response dynamics were too rapid (<100ms) for top-down attention signals to have developed. We cite a few studies suggesting attention cannot be evoked in tens of milliseconds. These factors argue against attentional processing of our stimuli and are now mentioned in the Results, Discussion, and Materials and methods sections - the dynamics are likely the result image-driven visual processing and not endogenous arousal or attention.

Third, none of the PIT examples in Figure 2 showed preferred face configuration over non-face configurations in the early response window as Figure 3's graphs indicated and the text suggested. The reversal of neuronal preference from face in early responses to non-face in late responses was supposed to be key evidence in support of the idea of the error signals, and should be demonstrated in Figure 2. These concerns should be addressable by revising the claims appropriately, adding cautionary notes/caveats in the interpretation of the data, or maybe providing some additional analysis, graphs and clarification.

We did not intend to imply that neurons reverse selectivity and attempted to avoid using the reversal language in our initial submission and have further refined our current resubmission. Rather, we simply show that a subset of neurons responded more strongly to the altered face part configurations in the late phase. We have added a statement to the Results section clarifying this main data claim.